# An Energy Balance Model for Paleoclimate Transitions

Brady Dortmans, William F. Langford, and Allan R. Willms

Department of Mathematics and Statistics, University of Guelph, 50 Stone Road West, Guelph, ON, Canada N1G 2W1

*Correspondence to:* W. F. Langford (wlangfor@uoguelph.ca)

**Abstract.** A new energy balance model (EBM) is presented and is used to study Paleoclimate transitions. While most previous EBMs dealt only with the globally averaged climate, this new EBM has three variants: Arctic, Antarctic and Tropical climates. The EBM incorporates the greenhouse warming effects of both carbon dioxide and water vapour, and also includes ice-albedo feedback and evapotranspiration. The main conclusion to be inferred from this EBM is that the climate system may possess
multiple equilibrium states, both warm and frozen, which coexist mathematically. While the actual climate can exist in only one of these states at any given time, the EBM suggests that climate can undergo transitions between the states, via mathematical saddlenode bifurcations. This paper proposes that such bifurcations have actually occurred in Paleoclimate transitions. The EBM is applied to the study of the *Pliocene Paradox*, the *Glaciation of Antarctica* and the so-called *warm, equable climate problem* of both the mid-Cretaceous Period and the Eocene Epoch. In all cases, the EBM is in qualitative agreement with the
geological record.

*Copyright statement.* TEXT

## 1 Introduction

For approximately 75% of the last 540 million years of the paleoclimate history of the Earth, the climate of both polar regions was mild and free of permanent ice-caps (Cronin, 2010; Crowley, 2000; Hubert et al., 2000). Today, both North and South
Poles are ice-capped; however, there is overwhelming evidence that these polar ice-caps are now melting. The Arctic is warming faster than any other region on Earth. The formation of the present-day Arctic and Antarctic ice-caps occurred abruptly, at widely separated times in the geological history of the Earth. This paper explores some of the the underlying mechanisms and forcing factors that have caused climate transitions in the past, with a focus on transitions that were *abrupt*. The primary objective is an exploration of the relevance of mathematical *bifurcation theory* in explaining these abrupt transitions. The un-
derstanding gained here will be applied in a subsequent paper to the important problem of determining whether anthropogenic climate change may lead to an abrupt transition in the future.

We present a new two-layer energy balance model (EBM) for the climate of the Earth. General knowledge of climate and of climate change has been advanced by many studies employing simple EBMs (Budyko, 1968; Kaper and Engler, 2013; McGehee and Lehman, 2012; North et al., 1981; Payne et al., 2015; Sagan and Mullen, 1972; Sellers, 1969; Stap et al., 2017).

In general, these EBMs facilitate exploration of the relationship between specific climate forcing mechanisms and the resulting climate changes. The EBM presented here includes a more accurate representation of the role of greenhouse gases in climate change than has been the case for previous EBMs. The model is based on fundamental principles of atmospheric physics, such as the Beer-Lambert Law, the Stefan-Boltzmann Law, the Clausius-Clapeyron equation and the ideal gas equation. In

particular, the modelling of water vapour acting as a greenhouse gas in the atmosphere, presented in Subsection 2.3.3, is more physically accurate that in previous EBMs and it shows that *water vapour feedback* is important in climate change. Also, *ice-albedo feedback* plays a central role in this EBM. The nonlinearity of this EBM leads to *bistability* (existence of multiple stable equilibrium states), to *hysteresis* (the climate state realized in the model depends on the past history) and to *bifurcations* (abrupt transitions from one state to another). Forcing factors that are included explicitly in the model include insolation, $CO_2$

concentration, relative humidity, evapotranspiration, ocean heat transport and atmospheric heat transport. Other factors that may affect climate change, such as geography, precipitation, vegetation, and continental drift are not included explicitly in the EBM, but are present only implicitly in so far as they affect those included factors.

During the Pliocene Epoch, 2.6–5.3 Ma, the climate of the Arctic region of Earth changed abruptly from ice-free to ice-capped. The major climate forcing factors (solar constant, orbital parameters, $CO_2$ concentration and locations of the conti-

nents) were all very similar to today (Fedorov et al., 2006, 2010; Haywood et al., 2009; Lawrence et al., 2009; De Schepper et al., 2015). Therefore, it is difficult to explain why the early Pliocene climate was so different from that of today. That problem has been called the *Pliocene Paradox* (Cronin, 2010; Fedorov et al., 2006, 2010). Currently, there is great interest in mid-Pliocene climate as a natural analogue of the future warmer climate expected for Earth, due to anthropogenic forcing. In recent years, significant progress has been achieved in understanding Pliocene climate, for example by Haywood et al. (2009);

Salzmann et al. (2009); Ballantyne et al. (2010); Steph et al. (2010); von der Heydt and Dijkstra (2011); Zhang and Yan (2012); Zhang et al. (2013); Sun et al. (2013); De Schepper et al. (2013); Tan et al. (2017); Chandan and Peltier (2017, 2018); Fletcher et al. (2018); see also references therein. This paper proposes that bistability and bifurcation may have played a fundamental role in determining the Pliocene climate.

The waning of the warm ice-free paleoclimate at the South Pole, leading eventually to abrupt glaciation of Antarctica at the

Eocene-Oligocene Transition (EOT) about 34 Ma, is believed to have been caused primarily by two major geological changes (although other factors played a role). One is the movement of the Continent of Antarctica from the Southern Pacific Ocean to its present position over the South Pole, followed by the development of the Antarctic Circumpolar Current (ACC), thus drastically reducing ocean heat transport to the South Pole (Cronin, 2010; Scher et al., 2015). The second is the gradual draw-down in $CO_2$ concentration world-wide (DeConto et al., 2008; Goldner et al., 2014; Pagani et al., 2011). The EBM of this

paper includes both $CO_2$ concentration and ocean heat transport as explicit forcing factors. The results of this paper suggest that the abrupt glaciation of Antarctica at the EOT was the result of a *bifurcation* that occurred as both of these factors changed incrementally; and furthermore, that both are required to explain the abrupt onset of Antarctic glaciation.

In the mid-Cretaceous Period (100 Ma) the climate of the entire Earth was much more *equable* than it is today. This means that, compared to today, the pole-to-equator temperature gradient was much smaller and also the summer/winter variation in

temperature at mid to high latitudes was much less (Barron, 1983). The differences in forcing factors between the Cretaceous

Period and modern times appeared to be insufficient to explain this difference in the climates. Eric Barron called this *the warm, equable Cretaceous climate problem* and he explored this problem in a series of pioneering papers (Barron et al., 1981; Barron, 1983; Sloan and Barron, 1992; Barron et al., 1995). In a similar vein, Huber and Caballero (2011) and Sloan and Barron (1990, 1992) observed that the early Eocene (56–48 Ma) encompasses the warmest climates of the past 65 million years, yet climate modelling studies had difficulty explaining such warm and equable temperatures. Therefore, this situation was called the *early Eocene warm equable climate problem*. The EBM of the present paper suggests that the mathematical mechanism of *bistability* provides plausible answers to both the mid-Cretaceous and the early Eocene equable climate problems. In fact, from the perspective of this simple model, these two are mathematically the same problem. Therefore, we call them collectively the *warm equable Paleoclimate problem*. Recent progress in paleoclimate science has succeeded in narrowing the gap between proxy and GCM estimates for the Cretaceous climate (Donnnadieu et al., 2006; Ladant and Donnadieu, 2016; O'Brien et al., 2017) and for the Eocene climate (Baatsen et al., 2018; Hutchinson et al., 2018; Lunt et al., 2016, 2017).

The principal contribution of this paper is a simple climate EBM, based on fundamental physical laws, that exhibits *bistability*, *hysteresis* and *bifurcations*. We propose that these three phenomena have occurred in the paleoclimate record of the Earth and they help to explain certain paleoclimate transitions and puzzles as outlined above. A key property of this EBM is that its underlying physical principles are highly *nonlinear*. As is well known, nonlinear equations can have multiple solutions, unlike linear equations, which can have only one unique solution (if well-posed). In our EBM, the same set of equations can have two or more co-existing stable solutions (bistability), for example an ice-capped solution (like today's climate) and an ice-free solution (like the Cretaceous climate), even with the same values of the forcing parameters. The determination of which solution is actually realized by the planet at a given time is dependent on past history (hysteresis). Changes in forcing parameters may drive the system abruptly from one stable state to another, at so-called "tipping points". In this paper, these tipping points are investigated mathematically, and are shown to be *bifurcation points*, which are investigated using mathematical bifurcation theory. Bifurcation theory tells us that the existence of bifurcation points is preserved (but the numerical values may change) under small deformations of the model equations. Thus, even though this conceptual model may not give us precise quantitative information about climate changes, qualitatively there is good reason to believe that the *existence* of the bifurcation points in the model will be preserved in similar more refined models and in the real world.

Previous climate models exhibiting multiple equilibrium states have indicated that bifurcations may cause abrupt climate transitions (Budyko, 1968; Sellers, 1969; North et al., 1981; Paillard, 1998; Alley et al., 2003; Rial et al., 2004; Lindsay and Zhang, 2005; Ferreira et al., 2010; Thorndike, 2012; Payne et al., 2015; Stap et al., 2017). None of these authors employ the recently developed mathematical theory of bifurcations to the extent used in this paper. Abrupt transitions in Quaternary glaciations have been studied by Ganopolski and Rahmstorf (2001); Paillard (2001a, b); Calov and Ganopolsky (2005) and Robinson et al. (2012). These glacial cycles are strongly influenced by orbital forcings, which are not within the scope of this paper. Scientists have long considered that abrupt transitions (or bifurcations) in the Atlantic Meridional Overturning Circulation (AMOC) were possible, and that this could contribute to abrupt climate change (Stommel, 1961; Rahmstorf, 1995; Ganopolski and Rahmstorf, 2001; Lenton et al., 2012). This type of change in the AMOC is sometimes called a *Stommel*

*bifurcation*. However, that phenomenon also is outside the scope of the EBM of this paper, which does not include ocean geography or meridional dependence. (An extension of this EBM, to a PDE spherical shell model, is planned.)

With regard to the Eocene-Oligocene transition (EOT), a variety of indicators, including the analysis of fossil plant stomata (Steinthorsdottir et al., 2016), imply that decreasing atmospheric $pCO_2$ in the Eocene preceded the large shift in oxygen isotope records that characterize the EOT. It was hypothesized in Steinthorsdottir et al. (2016) that at the EOT, a certain threshold of $pCO_2$ was crossed, resulting in an abrupt change of climate mode. Possible mechanisms for the threshold that they hypothesized are studied in Section 3.2. Even for changes as recent as the mid-Holocene, there is a debate, for example, over whether the abrupt desertification of the Sahara is due to a bifurcation, as suggested in Claussen et al. (1999) using an Earth Model of Intermediate Complexity (EMIC), or is a transient response of the AMOC to a sudden termination of freshwater discharge to the North Atlantic, as proposed in Liu et al. (2009), using a coupled Atmosphere-Ocean GCM. Similarly, for the glaciation of Greenland at the Pliocene-Plistocene transition, recent work in Tan et al. (2018), using a coupled GCM - ice sheet model, shows good agreement with proxy records without the need for bifurcations. The model of this paper does not adapt to these two situations, which are localized and away from the poles where the axis of symmetry restricts the dynamics and facilitates the analysis presented here.

Further investigation of climate changes, using a range of climate models from EBM and EMIC to GCM and OAGCM is warranted, to clarify the underlying mechanisms of abrupt climate change. Very sophisticated GCM, which include many 3D processes, are only able to run a few climate trajectories; while EBM and EMIC may explore more possibilities and investigate climate transitions (tipping points) with major simplifications and with less effort. More rigorous mathematical analysis is possible on small models, which may then suggest lines of inquiry on large models. The Earth climate system is extremely complex. For best results, a hierarchy of climate models is necessary.

The energy balance model presented here is conceptual and qualitative. It contains many simplifying assumptions and is not intended to give a detailed description of the climate of the Earth with quantitative precision. It complements but does not replace more detailed General Circulation Models (GCM). Geographically, this model is as simple as possible. It follows in a long tradition of *slab models* of the atmosphere. Previous slab models represented the atmosphere of the Earth as a globally-averaged uniform slab at a single temperature $T$. The temperature $T$ is determined in those models by a global energy balance equation of the form *energy in = energy out*. Such models are unable to differentiate between different climates at different latitudes; for example, if the polar climate is changing more rapidly that the tropical climate. In this new model, the forcing parameters of the slab atmosphere are chosen to represent each one of three particular latitudes: Arctic, Antarctic or Tropics. Each of these regions is represented by its own slab model, with its own forcing parameters and its own surface temperature $T_S$. In addition, each region has its own variable $I_A$ representing the intensity of the radiation re-emitted by the atmosphere. The two independent variables $I_A$ and $T_S$ are determined in each model by two energy balance equations, expressing energy balance in the atmosphere and energy balance at the surface, respectively. In this way, the different climate responses of these three regions to their respective forcings can be explored.

The role played by greenhouse gases in climate change is a particular focus of this model. Greenhouse gases trap heat emitted by the surface and are major contributors to global warming. The very different roles of the two principal greenhouse

gases in the atmosphere, carbon dioxide and water vapour, are analyzed here in Sections 2.3.2 and 2.3.3, respectively. The greenhouse warming effect of $CO_2$ increases with the density of the atmosphere but is independent of temperature, while the greenhouse warming of $H_2O$ increases with temperature but is independent of the density (or partial pressure) of the other gases present. The greenhouse warming of methane ($CH_4$) acts in a similar fashion to that of $CO_2$; therefore, $CH_4$ can be incorporated in the $CO_2$ concentration. As an increase in $CO_2$ concentration causes climate warming, this warming causes an increase in evaporation of $H_2O$ into the atmosphere, which further increases the climate warming beyond that due to $CO_2$ alone (this is true both in the model and in the real atmosphere). This effect is known as *water vapour feedback*. The energy balance model presented here is the first EBM to incorporate these important roles of the greenhouse gases in such detail.

The paper concludes with two Appendices. In Appendix A, model parameters that are difficult or impossible to determine for paleoclimates are calibrated using the abundant satellite and surface data available for today's climate. In addition, justification is given for parameter values chosen for the model. In Appendix B the paleoclimate model of this paper is adapted to modern day conditions and its equilibrium climate sensitivity (ECS) s determined. Here, ECS is the change in global mean temperature produced by a doubling of $CO_2$ in the model, starting from the pre-industrial value of 270 ppm. For this EBM, the ECS is determined to be $\Delta T = 3.3°C$, which is at the high end of the range accepted by the IPCC (IPCC, 2013).

## 2   The Energy Balance Climate Model

In this energy balance model (EBM), the atmosphere and surface are each assumed to be in energy balance. Short-wave radiant energy from the sun is partly reflected by the atmosphere back into space, a small portion is absorbed directly by the atmosphere, and the remainder passes through to the surface. The surface reflects some of this short-wave energy (which is assumed to escape to space) and absorbs the rest, re-emitting long wave radiant energy of intensity $I_S$, upward into the atmosphere. The atmosphere is modelled as a slab, with greenhouse gases, that absorbs a fraction $\eta$ of the radiant energy $I_S$ from the surface. The atmosphere re-emits radiant energy of total intensity $I_A$. Of this radiation $I_A$, a fraction $\beta$ is directed downward to the surface, and the remaining fraction $(1 - \beta)$ goes upward and escapes to space.

This model is based on the uniform slab EBM used in Payne et al. (2015), modified as shown in Figure 1. In our case, the "slab" is a uniform column of air, of unit cross-section, extending vertically above the surface to the tropopause, and located either at a pole or at the equator. The symbols in Figure 1 are defined in Table 1.

This section presents the mathematical derivation of the EBM. Readers interested only in the climate applications of the model may skip this Section and go directly to Section 3. A preliminary version of this EBM was presented in a conference proceedings paper (Dortmans et al., 2018). The present model incorporates several important improvements over that model. The differences between that model and the one presented here are indicated where appropriate in the text. Furthermore, the previous paper (Dortmans et al., 2018) considered only the application of the model to the Arctic climate and the Pliocene Paradox; it did not study Antarctic or Tropical climate or the Cretaceous warm equable climate problem, as does the present paper.

| Variables and Parameters Used | | |
|---|---|---|
| **Variables** | **Symbol** | **Values** |
| Mean temperature of the surface | $T_S$ | -50 to +20 C |
| Infrared radiation from the surface | $I_S$ | 141 to 419 W m$^{-2}$ |
| Mean temperature of the atmosphere | $T_A$ | -70 to 0 C |
| Energy emitted by the atmosphere | $I_A$ | 87 to 219 W m$^{-2}$ |
| **Parameters and Constants** | **Symbol** | **Values** |
| Temperature of freezing point for water | $T_R$ | 273.15 K |
| Stefan-Boltzmann constant | $\sigma$ | 5.670 10$^{-8}$ W m$^{-2}$ K$^{-4}$ |
| Emissivity of dry air | $\epsilon$ | 0.9 |
| Fraction of $I_A$ reaching surface | $\beta$ | 0.63 |
| Incident solar radiation | $Q$ | 173.2 W m$^{-2}$ (poles), 418.8 W m$^{-2}$ (equator) |
| Fraction of solar radiation reflected from atmosphere | $\xi_R$ | 0.2235 |
| Fraction of solar radiation directly absorbed by atmosphere | $\xi_A$ | 0.2324 |
| Warm surface albedo | $\alpha_W$ | 0.08 to 0.15 |
| Cold surface albedo | $\alpha_C$ | 0.7 |
| Albedo transition rate (in tanh function) | $\omega = \Omega/T_R$ | 0.01 |
| Solar radiation striking surface | $F_S$ | $(1 - \xi_R - \xi_A)Q$ |
| Ocean heat transport | $F_O$ | -55 to 100 W m$^{-2}$ |
| Atmospheric heat transport | $F_A$ | -25 to 45 W m$^{-2}$ |
| Vertical heat transport (conduction + evapotranspiration) | $F_C$ | 0 to 150 W m$^{-2}$, function of $T_S$ |
| Vertical heat transport coefficients | $a_1, a_2$ | 2.650 and 6.590 × 10$^{-2}$, respectively |
| Molar concentration of CO$_2$ in ppm | $\mu$ | 270 to 1600 ppm |
| Relative humidity of H$_2$O | $\delta$ | 0.5 to 0.85 |
| Absorptivity for CO$_2$ | $\eta_C$ | 0 to 1, function of $\mu$ |
| Absorptivity for H$_2$O | $\eta_W$ | 0 to 1, function of $\delta$ and $T_S$ |
| Absorptivity for clouds | $\eta_{Cl}$ | 0.3729 |
| Total atmosphere absorptivity | $\eta$ | $1 - (1 - \eta_W)(1 - \eta_C)(1 - \eta_{Cl})$ |
| Grey gas absorption coefficient for CO$_2$ | $k_C$ | 0.07424 m$^2$ kg$^{-1}$ |
| Grey gas absorption coefficient for H$_2$O | $k_W$ | 0.05905 m$^2$ kg$^{-1}$ |
| Standard and normalized atmosphere lapse rates | $\Gamma, \gamma = \Gamma/T_R$ | 6.49×10$^{-3}$ K m$^{-1}$ and 2.38×10$^{-5}$ m$^{-1}$, respectively |
| Tropopause height | $Z$ | 9 km (poles), 17 km (equator), 14 km (global) |
| Latent heat of vaporization of water | $L_v$ | 2.2558 10$^6$ m$^2$ s$^{-2}$ |
| Ideal gas constant specific to water vapour | $R_W$ | 461.5 m$^2$ s$^{-2}$ K$^{-1}$ |
| Saturated partial pressure of water at $T_R$ | $P_W^{sat}(T_R)$ | 611.2 Pa |
| Atmospheric standard pressure at surface | $P_A$ | 101.3 × 10$^3$ Pa |
| Gravitational acceleration | $g$ | 9.81 m s$^{-2}$ |
| Parameter for $\eta_C$ | $G_C$ | $(1.52 \times 10^{-6})k_C P_A/g = 1.166 \times 10^{-3}$ |
| First parameter for $\eta_W$ | $G_{W1}$ | $L_v/(R_W T_R) = 17.89$ |
| Second parameter for $\eta_W$ | $G_{W2}$ | $k_W P_W^{sat}(T_R)/(\gamma R_W T_R) = 12.05$ |

**Table 1.** Summary of variables and parameters used in the model. Many of these parameters have standard textbook values (e.g. $T_R, \sigma, L_v, R_W$). The standard lapse rate $\Gamma$ is from ICAO (1993). Other parameters such as $k_C$ and $k_W$ are derived in Appendix A.

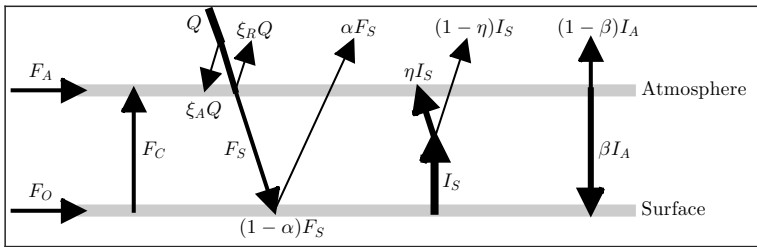

**Figure 1.** A visualization of the energy balance model. Symbols are defined in Table 1 and Section 2.1.

## 2.1 Energy Balance

The model consists of two energy balance equations, one for the atmosphere and one for the surface. In Figure 1, the so-called *forcings* are shown as arrows, pointing in the direction of energy transfer. From Figure 1, the energy balance equations for the atmosphere and surface are given respectively by

$$0 = F_A + F_C + \xi_A Q + \eta I_S - I_A, \quad \text{and} \tag{1}$$

$$0 = F_O - F_C + (1 - \alpha)F_S - I_S + \beta I_A. \tag{2}$$

Symbols and parameter values for the model are defined in Table 1. Appendix A4 provides derivations and justification for the values of the empirical parameters. The forcings $F_O$ and $F_A$ represent ocean and atmosphere heat transport, respectively, and are specified as constants for each region of interest. Heat transport by conduction/convection from the surface to the atmosphere is denoted $F_C$. This quantity will be largely dependent on surface temperature, $T_S$. As described in Appendix A we have modelled it as an hyperbola that is mostly flat for temperatures below freezing, and grows roughly linearly for temperatures above freezing so that

$$F_C = A_1 (T_S - T_R) + \sqrt{A_1^2(T_S - T_R)^2 + A_2^2}, \tag{3}$$

where $A_1$ and $A_2$ are constants. Since the model is concerned with temperatures around the freezing point of water, we set this as a reference temperature, $T_R = 273.15 \, \text{K}$.

The annually averaged intensity of solar radiation striking a surface parallel to the earth's surface but at the top of the atmosphere is $Q$. The value of $Q$ at either Pole is $Q = 173.2 \, \text{Wm}^{-2}$ and at the Equator is $418.8 \, \text{Wm}^{-2}$ (McGehee and Lehman, 2012; Kaper and Engler, 2013). A fraction $\xi_R$ of this short wave radiation is reflected by the atmosphere back into space and a further fraction $\xi_A$ is directly absorbed by the atmosphere; the remainder penetrates to the surface. See Appendix A1 for the derivation of values for $\xi_R$ and $\xi_A$. The solar radiation striking the surface of the Earth is

$$F_S = (1 - \xi_R - \xi_A)Q. \tag{4}$$

The surface *albedo* is the fraction, $\alpha$, of this solar radiation that reflects off the surface back into space. Thus the solar forcing absorbed by the surface is $(1-\alpha)F_S$, and the solar radiation reflected back to space is $\alpha F_S$. Typical values of the surface albedo

$\alpha$ are $0.6 - -0.9$ for snow, $0.4 - -0.7$ for ice, $0.2$ for crop land and $0.1$ or less for open ocean. In this paper we introduce a smoothly varying albedo given by the hyperbolic tangent function:

$$\alpha = \frac{1}{2}\left([\alpha_W + \alpha_C] + [\alpha_W - \alpha_C]\tanh\left(\frac{T_S - T_R}{\Omega}\right)\right), \tag{5}$$

where $\alpha_C$ and $\alpha_W$ are the albedo values for cold and warm temperatures, respectively, and the parameter $\Omega$ determines the steepness of the transition between $\alpha_C$ and $\alpha_W$. See Appendix A2 for a full explanation of Equation (5).

The emission of long wave radiation, $I$, from from a body is governed by the *Stefan-Boltzmann law*, $I = \epsilon\sigma T^4$, where $\epsilon$ is the emissivity, $\sigma$ is the Stefan-Boltzmann constant, and $T$ is temperature. The surface of the Earth acts as a black-body, so for the radiation emitted from the surface $\epsilon = 1$, thus

$$I_S = \sigma T_S^4. \tag{6}$$

Previous authors have postulated an idealized uniform atmospheric temperature $T_A$ for the slab model, so that the intensity of radiation emitted by the atmosphere, $I_A$, is

$$I_A = \epsilon\sigma T_A^4. \tag{7}$$

The emissivity is $\epsilon = 0.9$ since the atmosphere is an imperfect black-body. A uniform temperature for the atmosphere does not exist in the real world, where $T_A$ varies strongly with height, unlike $T_S$, which has a single value. Previous two-layer EBMs have used $(T_S, T_A)$ as the two independent variables in the two energy balance equations (1) and (2). Here instead, we use $(T_S, I_A)$ as the two independent variables, and then we formally let $T_A$ be *defined* by (7). The fraction of $I_A$ that reaches the surface is $\beta = 0.63$; see Appendix A6.

The parameter $\eta$ represents the fraction of the infrared radiation $I_S$ from the surface that is absorbed by the atmosphere and is called *absorptivity*. The major constituents of the atmosphere are nitrogen and oxygen and these gases do not absorb any infrared radiation. The gases that do contribute to the absorptivity $\eta$ are called *greenhouse gases*. Chief among these are carbon dioxide and water vapour. The contribution of these two greenhouse gases to $\eta$ are analyzed in Sections 2.3.2 and 2.3.3, respectively. Although both contribute to warming of the climate, the underlying physical mechanisms of the two are very different. In general, the contribution to $\eta$ of water vapour is a function of temperature. Another major contributor to absorption is the liquid and solid water in clouds. We model this portion of the absorption as constant, since we do not include any data on cloud cover variation. However, we experimented with making this portion vary with surface temperature, and the results were not qualitatively different than those presented here.

### 2.1.1 Nondimensional Temperatures

We rescale temperature by the reference temperature $T_R = 273.15\,\text{K}$ and define new nondimensional temperatures and new parameters

$$\tau_S = \frac{T_S}{T_R}, \quad i_A = \frac{I_A}{\sigma T_R^4}, \quad q = \frac{Q}{\sigma T_R^4}, \quad f_O = \frac{F_O}{\sigma T_R^4}, \quad f_A = \frac{F_A}{\sigma T_R^4}, \quad f_C = \frac{F_C}{\sigma T_R^4}, \quad a_1 = \frac{A_1}{\sigma T_R^3}, \quad a_2 = \frac{A_2}{\sigma T_R^4}, \quad \omega = \frac{\Omega}{T_R}. \tag{8}$$

After normalization, the freezing temperature of water is represented by $\tau = 1$ and the atmosphere and surface energy balance equations (1)–(6) simplify to

$$i_A = f_A + f_C(\tau_S) + \xi_A q + \eta(\tau_S)\tau_S^4, \tag{9}$$

$$i_A = \frac{1}{\beta}\left[f_C(\tau_S) - f_O - (1 - \alpha(\tau_S))(1 - \xi_R - \xi_A)q + \tau_S^4\right], \tag{10}$$

where, from Appendix A

$$f_C(\tau_S) = a_1(\tau_S - 1) + \sqrt{a_1^2(\tau_S - 1)^2 + a_2^2}, \tag{11}$$

$$\alpha(\tau_S) = \frac{1}{2}\left([\alpha_W + \alpha_C] + [\alpha_W - \alpha_C]\tanh\left(\frac{\tau_S - 1}{\omega}\right)\right). \tag{12}$$

The range of surface temperatures $T_S$ observed on Earth is restricted to an interval around the freezing point of water, 273.15 K, and therefore the nondimensional temperature $\tau$ lies in an interval around $\tau = 1$. In this paper, we assume $0.8 \leq \tau \leq 1.2$, which corresponds approximately to a range in more familiar Celsius degrees of $-54°C \leq T \leq +54°C$. Another reason for the upper limit on temperature is that the Clausius-Clapeyron Law used in Section 2.3.3 fails to apply at temperatures above the boiling point of water.

## 2.2 Optical Depth and the Beer-Lambert Law

The goal of this section is to define the absorptivity parameter $\eta$ in the EBM Equation (9) (or (1)), in such a way that the atmosphere in the uniform slab model will absorb the same fraction $\eta$ of the longwave radiation $I_S$ from the surface as does the real nonuniform atmosphere of the Earth. This absorption is due primarily to water vapour, carbon dioxide, and clouds. Previous energy balance models have assigned a constant value to $\eta$, often determined by climate data. In the present EBM, $\eta$ is not constant but is a function of other more fundamental physical quantities, such as $\mu, \delta, k_C, k_W$ and $T$. This function is determined by classical physical laws. In this way, the present EBM adjusts automatically to changes in these physical quantities, and represents a major advance over previous EBMs.

The Beer-Lambert Law states that when a beam of radiation (or light) enters a sample of absorbing material, the absorption of radiation at any point $z$ is proportional to the intensity of the radiation $I(z)$ and also to the concentration or density of the absorber $\rho(z)$. This bilinearity fails to hold at very high intensity of radiation or high density of absorber, neither of which is the case in the Earth's atmosphere. Whether this Law is applied to the uniform slab model or to the nonuniform real atmosphere, it yields the same differential equation

$$\frac{dI}{dz} = -k\rho(z)I(z), \tag{13}$$

where $k$ $(m^2\,kg^{-1})$ is the absorption coefficient of the material, $\rho$ $(kg\,m^{-3})$ is the density of the absorbing substance such as $CO_2$, and $z$ $(m)$ is distance along the path. The differential Equation (13) may be integrated from $z = 0$ (the surface) to $z = Z$ (the tropopause), to give

$$\frac{I_T}{I_S} = e^{-\int_0^z k\rho(z)\,dz} \equiv e^{-\lambda}, \quad \text{where} \quad \lambda \equiv \int_0^Z k\rho(z)\,dz. \tag{14}$$

Here $I_T \equiv I(Z)$ is the intensity of radiation escaping to space at the Tropopause $z = Z$, and $\lambda$ is the so-called *optical depth* of the material. Note that $\lambda$ is dimensionless. The absorptivity parameter $\eta$ in Equations (1) and (9), represents the fraction of the outgoing radiation $I_S$ from the surface that is absorbed by the atmosphere (not to be confused with the absorption coefficient $k$). It follows from the Beer-Lambert Law that $\eta$ is completely determined by the corresponding optical depth parameter $\lambda$; that is

$$\eta = \frac{I_S - I_T}{I_S} = 1 - e^{-\lambda}. \tag{15}$$

For a mixture of $n$ attenuating materials, with densities $\rho_i$, absorption coefficients $k_i$ and corresponding optical depths $\lambda_i$, the Beer-Lambert Law extends to

$$\frac{I_T}{I_S} = e^{-\Sigma_{i=1}^n \int_0^Z k_i \rho_i(z)\, dz} \equiv e^{-\Sigma_{i=1}^n \lambda_i}, \tag{16}$$

so that

$$\eta = 1 - e^{-\Sigma_{i=1}^n \lambda_i} = 1 - \prod_{i=1}^n e^{-\lambda_i} = 1 - \prod_{i=1}^n (1 - \eta_i), \qquad \text{where} \qquad \eta_i = 1 - e^{-\lambda_i}. \tag{17}$$

Equation (17) is the key to solving the problem posed in the first sentence of this subsection. For the $i^{th}$ absorbing material in the slab model, we set its optical depth $\lambda_i$ to be equal to the value of the optical depth that this gas has in the Earth's atmosphere as given by (14), and then combine them using (16). This calculation is presented for the case of $CO_2$ in Subsection 2.3.2 and for water vapour in 2.3.3. For the third absorbing material in our model, clouds, we assume a constant value $\eta_{Cl}$.

## 2.3 Greenhouse Gases

The two principal greenhouse gases are carbon dioxide $CO_2$ and water vapour $H_2O$. Because they act in different ways, we determine the absorptivities $\eta_C$, $\eta_W$ and optical depths $\lambda_C$, $\lambda_W$ of $CO_2$ and $H_2O$ separately, and then combine their effects, along with the absorption due to clouds, $\eta_{Cl}$, using the Beer-Lambert Law for mixtures, Equation (17). Methane acts similarly to $CO_2$ and can be included in the optical depth for $CO_2$. Other greenhouse gases have only minor influence and are ignored in this paper.

### 2.3.1 The Grey Gas Approximation

Although it is well-known that gases like $CO_2$ and $H_2O$ absorb infrared radiation $I_S$ only at specific wavelengths (spectral lines), in this paper the *grey gas approximation* is used; that is, the absorption coefficient $k_C$ or $k_W$ is given as a single number averaged over the infrared spectrum (Pierrehumbert, 2010). The thesis Dortmans (2017) presents a survey of values in the literature for the absorption coefficients $k_C$ and $k_W$ of $CO_2$ and $H_2O$, respectively, in the grey gas approximation. The values used in this paper are given in Table 1; and are derived as described in Appendix A5.

### 2.3.2 Carbon Dioxide

The concentration of $CO_2$ in the atmosphere is usually expressed as a ratio, in molar parts per million (ppm) of $CO_2$ to dry air, and written as $\mu$. There is convincing evidence that $\mu$ has varied greatly in the geological history of the Earth, and has decreased slowly over the past 100 million years; however, today $\mu$ is increasing due to human activity. The value before the industrial revolution was $\mu = 270$ ppm, but today $\mu$ is slightly above 400 ppm.

Although traditionally $\mu$ is measured as a ratio of molar concentrations, in practice, both the density $\rho$ and the absorption coefficient $k$ are expressed in mass units of kg. Therefore, before proceeding, $\mu$ must be converted from a molar ratio to a ratio of masses in units of kg. The mass of one mole of $CO_2$ is approximately $mm_C = 44 \times 10^{-3}$ kg/mol. The dry atmosphere is a mixture of 78% Nitrogen, 21% Oxygen and 0.9% Argon, with molar masses of 28 g/mol, 32 g/mol and 40 g/mol, respectively. Neglecting other trace gases in the atmosphere, a weighted average gives the molar mass of the dry atmosphere as $mm_A = 29 \times 10^{-3}$ kg/mol. Therefore, the $CO_2$ concentration $\mu$ measured in molar ppm is converted to mass concentration in kg ppm by multiplication by the ratio $mm_C/mm_A \approx 1.52$. If $\rho_A(z)$ is the density of the atmosphere at altitude $z$ in kg/m$^3$, then the mass density of $CO_2$ at the same altitude, with molar concentration $\mu$ ppm, is

$$\rho_C(z) = 1.52 \frac{\mu}{10^6} \rho_A(z) \quad \text{kg/m}^3. \tag{18}$$

It is known that $CO_2$ disperses rapidly throughout the Earth's atmosphere, so that its concentration $\mu$ may be assumed independent of location and altitude (IPCC, 2013). As the density of the atmosphere decreases with altitude, the density of $CO_2$ decreases at exactly the same rate, according to (18). Substituting (18) into (14) determines the optical depth $\lambda_C$ of $CO_2$

$$\lambda_C = 1.52 \frac{\mu}{10^6} k_C \int_0^Z \rho_A(z)\,dz. \tag{19}$$

Now consider a vertical column of air, of unit cross-section, from surface to tropopause. The integral in (19) is precisely the total mass of this column. The atmospheric pressure at the surface is $P_A$, and this is the total weight of the column. Therefore $\int_0^Z \rho(A)\,dz = P_A/g$, where $g$ is acceleration due to gravity. Therefore, the optical depth $\lambda_C$ of $CO_2$ in the actual atmosphere in (19) is

$$\lambda_C = \mu G_C, \qquad \text{where} \qquad G_C \equiv 1.52 \times 10^{-6} k_C \frac{P_A}{g}. \tag{20}$$

With $\lambda_C$ so determined, it follows from (15) that

$$\eta_C = 1 - e^{-\lambda_C} = 1 - \exp(-\mu G_C). \tag{21}$$

As listed in Table 1 and derived in Appendix A5, the calibrated value for $k_C$ is 0.07424, and therefore the value for the greenhouse gas parameter for $CO_2$ is $G_C = 1.166 \times 10^{-3}$.

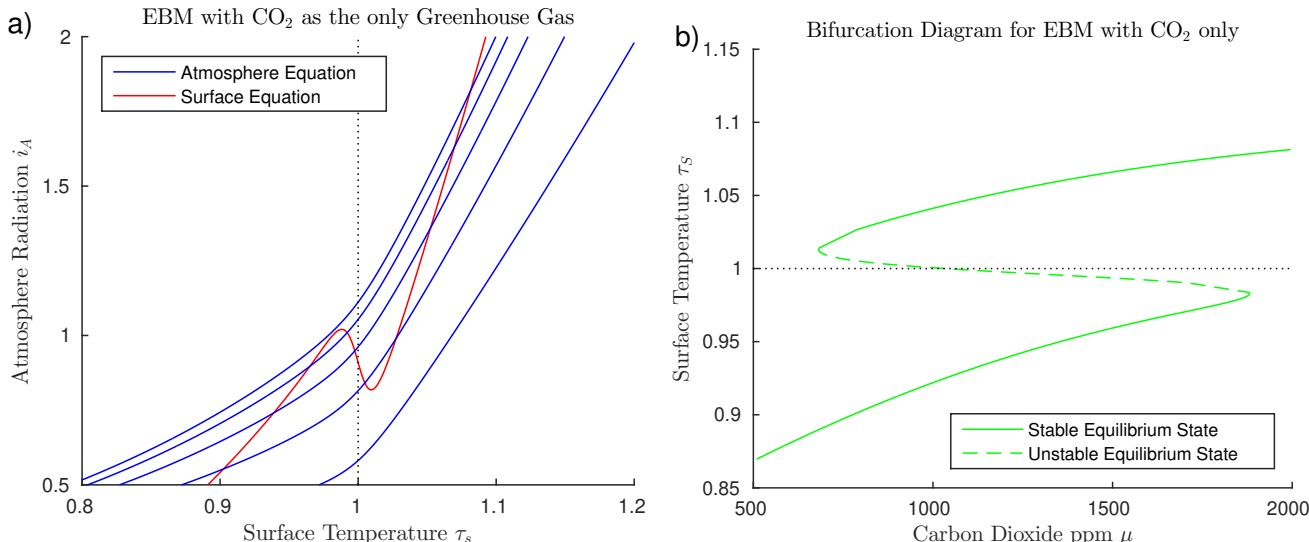

**Figure 2.** Dry atmosphere EBM (22)–(23) (with $F_A = 65 \ \mathrm{Wm}^{-2}$, $F_O = 50 \ \mathrm{Wm}^{-2}$, $Q = 173.2 \ \mathrm{Wm}^{-2}$, $\alpha_W = 0.08$, and $Z = 9$ km). a) $CO_2$ values $\mu = 400, 800, 1200, 1600, 2000$; from bottom to top blue curves. A unique cold equilibrium exists for $\mu = 400$ (bottom blue curve), multiple equilibria exist for $\mu = 800, 1200$, and $1600$, a unique warm equilibrium exists for $\mu = 2000$ (top blue curve). b) Bifurcation diagram for the same parameters as in a), showing the result of solving the EBM for $\tau_S$ as a function of the forcing parameter $\mu$, holding all other forcing parameters constant. Two saddlenode (or fold) bifurcations occur, at approximately $\mu = 681$ and $\mu = 1881$. Three distinct solutions exist between these two values of $\mu$. As $\mu$ decreases from right to left in this figure, if the surface temperature starts in the warms state ($\tau_S > 1$) then it abruptly "falls" from the warm state to the frozen state when the left bifurcation point is reached. Similarly, starting on the frozen state ($\tau_S < 1$), if $\mu$ increases sufficiently, then the state will jump upward to the warm branch when the right bifurcation point is reached. Both of these transitions are irreversible (one-way).

The *dry atmosphere EBM* is obtained by assuming there is no water vapour and no clouds in the atmosphere. Hence $\eta = \eta_C$ and $\delta = 0$. We assume also that $\xi_R = \xi_A = 0$. Making these substitutions in the EBM Equations (9)–(10) yields

$$i_A = f_A + f_C(\tau_S) + [1 - \exp(-\mu G_C)] \cdot \tau_S^4, \tag{22}$$

$$i_A = \frac{1}{\beta} \left( f_C(\tau_S) - f_O - [1 - \alpha(\tau_S)]q + \tau_S^4 \right). \tag{23}$$

5  Due to the nonlinearity of the ice-albedo function in (5), there can be in fact up to 3 points of intersection as shown in Figure 2a. If the $CO_2$ level $\mu$ decreases sufficiently the warm state ($\tau_S > 1$) disappears, while as $CO_2$ increases the frozen state ($\tau_S < 1$) may disappear. The $CO_2$ level $\mu$ is quite elevated in Figure 2, but the effects of water vapour as a greenhouse gas and of clouds have been ignored.

Figure 2b introduces a *bifurcation diagram* (Kuznetsov, 2004), in which the two EBM equations (22)–(23) have been solved

10  for the surface temperature $\tau_S$, which is then plotted as a function of the parameter $\mu$. Figure 2b shows the 3 distinct solutions $\tau_S$ of the EBM: a warm solution ($\tau_S > 1$), a frozen solution ($\tau_S < 1$), and a third solution that crosses through $\tau_S = 1$ and connects the other two solutions in *saddlenode bifurcations* (Kuznetsov, 2004). Stability analysis (Dortmans, 2017) shows that

the warm and frozen solutions are *stable* (in a dynamical systems sense), while the third solution (denoted by a dashed line) is unstable.

### 2.3.3 Water vapour and the Clausius-Clapeyron Equation

In this section, we determine the absorptivity of water vapour as a function of temperature, $\eta_W(\tau_S)$, using fundamental physical laws including the Clausius-Clapeyron Equation, the Ideal Gas Law, and the Beer-Lambert Law (Pierrehumbert, 2010). We also assume the idealized Lapse Rate of the International Standard Atmosphere (ISA) as defined by ICAO (ICAO, 1993). Unlike $CO_2$, the concentration of water vapour $H_2O$ in the atmosphere varies widely with location and altitude. This is because the partial pressure of $H_2O$ varies strongly with the local temperature. In fact, it is bounded by a maximum saturated value, which itself is a nonlinear function of temperature, $P_W^{sat}(T)$. The actual partial pressure of water vapour is then a fraction $\delta$ of this saturated value,

$$P_W(T) = \delta\, P_W^{sat}(T), \qquad 0 \le \delta \le 1, \tag{24}$$

where $\delta$ is called *relative humidity*. While $P_W^{sat}(T)$ varies greatly with $T$ in the atmosphere, the relative humidity $\delta$ is comparatively constant. When the actual $P_W(T)$ exceeds the saturated value $P_W^{sat}(T)$ (i.e. $\delta > 1$), the excess water vapour condenses out of the atmosphere and falls as rain or snow.

The saturated partial pressure at temperature $T$ is determined by the *Clausius-Clapeyron equation* (Pierrehumbert, 2010),

$$P_W^{sat}(T) = P_W^{sat}(T_R) \exp\left( \frac{L_v}{R_W} \left[ \frac{1}{T_R} - \frac{1}{T} \right] \right), \tag{25}$$

where $T_R$ is the reference temperature, here chosen to be the freezing point of water (273.15 K), $L_v$ is the latent heat of vaporization of water and $R_W$ is the ideal gas constant for water, see Table 1. The actual partial pressure of water vapour at relative humidity $\delta$ and temperature $T$ is then given by combining equations (24) and (25).

We may use the *Ideal Gas Law* in the form $P_W = \rho_W R_W T$ to convert the partial pressure $P_W$ of water vapour at temperature $T$ to mass density $\rho_W$ of water vapour at that temperature. Substituting into (24) and (25) gives

$$\rho_W(T) = \delta\, \frac{P_W^{sat}(T_R)}{R_W T} \exp\left( \frac{L_v}{R_W} \left[ \frac{1}{T_R} - \frac{1}{T} \right] \right). \tag{26}$$

Transforming to the dimensionless temperature $\tau = T/T_R$ as in Subsection 2.1.1, this becomes

$$\rho_W(\tau) = \delta \left( \frac{P_W^{sat}(T_R)}{R_W T_R} \right) \frac{1}{\tau} \exp\left( \frac{L_v}{R_W T_R} \left[ \frac{\tau - 1}{\tau} \right] \right). \tag{27}$$

The Beer-Lambert Law in Section 2.2 implies that the absorptivity of a greenhouse gas $\eta_i$ is completely determined by its optical depth $\lambda_i$. For water vapour, from Equations (15) and (14),

$$\eta_W = 1 - e^{-\lambda_W} \qquad \text{where} \qquad \lambda_W \equiv \int_0^Z k_W \rho_W(z)\, dz. \tag{28}$$

Here $k_W$ is the absorption coefficient of water vapour; see Appendix A5. In order to evaluate the integral in (28), we need to know how $\rho_W$ varies with height $z$. We have shown that $\rho_W$ is a function of temperature, given by (26) or (27). Therefore, we need an expression for the variation of temperature $T$ with height $z$. Under normal conditions, the temperature $T$ decreases with height in the troposphere. This rate of decrease is called the *lapse rate* $\Gamma$, and defined as

$$\Gamma \equiv -\frac{dT}{dz}. \tag{29}$$

Normally, $\Gamma$ is positive and is close to constant in value from the surface to the tropopause. The International Civil Aviation Organization has defined, for reference purposes, the *International Standard Atmosphere* (ISA), in which $\Gamma$ is assigned the constant value $\Gamma = 6.49 \times 10^{-3}$ K/m (ICAO, 1993). Using this assumption, the variation of temperature with height is given as

$$T(z) = T_S - \Gamma z \quad \text{or} \quad \tau(z) = \tau_S - \gamma z, \tag{30}$$

where the normalized lapse rate is $\gamma = \Gamma/T_R = 2.38 \times 10^{-5} \ m^{-1}$. The tropopause height $Z$ ranges from 8 to 11 km at the poles and 16 to 18 km at the equator, based on satellite measurements (Kishore et al., 2006). Therefore, both $T(Z)$ and $\tau(Z)$ are positive. For this paper we will take the height to be $Z = 9$ km at the poles and $Z = 17$ km at the equator.

Equation (30) may be used to change the variable of integration in Equation (28), for the optical depth of water vapour, from $z$ to $\tau$. The result is

$$\lambda_W = k_W \int_0^Z \rho_W(\tau(z)) \, dz = \frac{k_W}{\gamma} \int_{\tau_S - \gamma Z}^{\tau_S} \rho_W(\tau) \, d\tau. \tag{31}$$

Now substitute (27) into the integral (31) and simplify to

$$\lambda_W(\tau_S) = \delta \, G_{W2} \int_{\tau_S - \gamma Z}^{\tau_S} \frac{1}{\tau} \exp\left( G_{W1} \left[ \frac{\tau - 1}{\tau} \right] \right) d\tau, \tag{32}$$

where the greenhouse gas parameters $G_{W1}$ and $G_{W2}$ for water vapour are defined as

$$G_{W1} \equiv \frac{L_v}{R_W T_R} \quad \text{and} \quad G_{W2} \equiv \frac{k_W P_W^{sat}(T_R)}{R_W T_R \gamma}. \tag{33}$$

Finally, using the Beer-Lambert Law, the absorptivity $\eta_W$ of water vapour in Equation (28) is determined by its optical depth, and is now a function of the surface temperature,

$$\eta_W(\tau_S) = 1 - \exp[-\lambda_W(\tau_S)] = 1 - \exp\left[ -\delta \, G_{W2} \int_{\tau_S - \gamma Z}^{\tau_S} \frac{1}{\tau} \exp\left( G_{W1} \left[ \frac{\tau - 1}{\tau} \right] \right) d\tau \right]. \tag{34}$$

The definite integral in this expression is easily evaluated numerically in the process of solving the atmosphere and surface EBM equations. As given in Table 1 and derived in Appendix A5, the calibrated value of $k_W$ is 0.05905 m$^2$ kg$^{-1}$ and the greenhouse gas parameters for $H_2O$ are

$$G_{W1} = 17.89 \quad \text{and} \quad G_{W2} = 12.05.$$

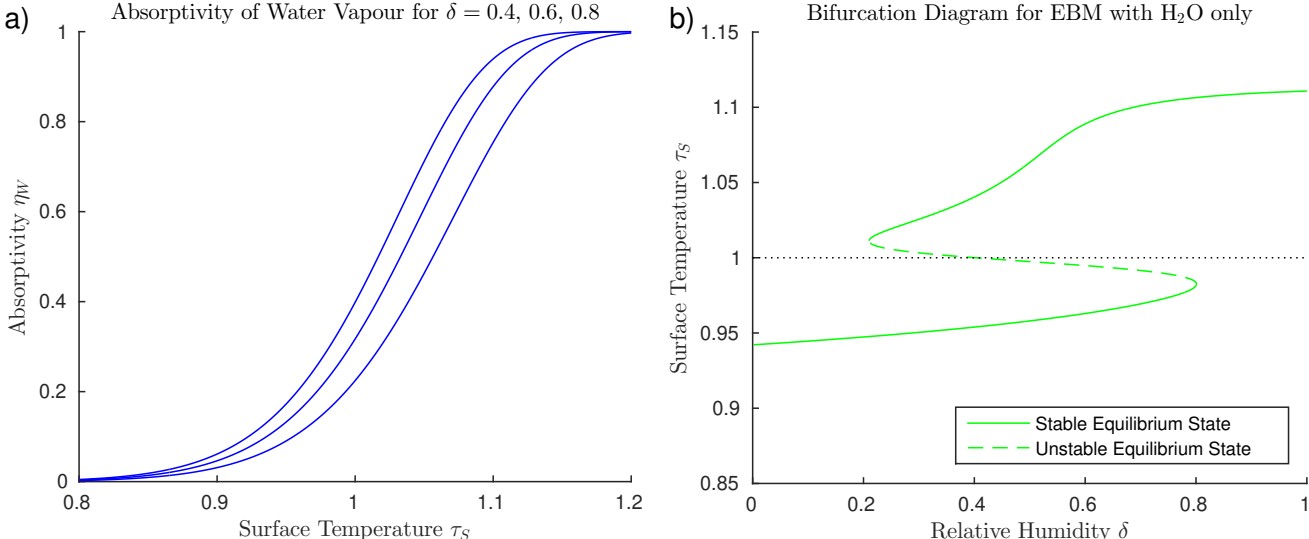

**Figure 3.** Climate EBM (9)–(10) including greenhouse effect of Water Vapour only, $\mu = 0$ (with $F_A = 45$ Wm$^{-2}$, $F_O = 50$ Wm$^{-2}$, $Q = 173.2$ Wm$^{-2}$, $\alpha_W = 0.08$, and $Z = 9$ km). a) Absorptivity $\eta_W(\tau_S)$ as given by Equation (34) with relative humidity $\delta = 0.4,\ 0.6,\ 0.8$, from bottom to top blue lines. Below freezing ($\tau_s < 1$) water vapour has little influence as a greenhouse gas, but absorptivity increases rapidly to approach $\eta = 1$ for $\tau > 1$. b) Bifurcation diagram for EBM with $\mu = 0$, and relative humidity increasing from 0 to 1. Note the accelerated increase in surface temperature $\tau_S$ above $\tau_S = 1$, compared to that in Figure 2b, due to the rapid increase in $\eta_W$.

Figure 3a shows that the function $\eta_W(\tau_S)$ in (34) increases rapidly from near 0 to near 1 as $\tau_S$ increases past 1, and it steepens towards a step function as $\delta$ increases. This implies that, if the surface temperature $\tau_S$ in the surface energy balance Equation (10) increases past $\tau_S = 1$, then the absorptivity of water vapour $\eta_W(\tau_S)$, acting as a greenhouse gas in Equation (9), increases rapidly, thus amplifying the heating of the atmosphere by the radiation $I_S = \sigma T_S^4$ from the surface. Energy balance

5 requires a corresponding increase in the radiation $I_A$ transmitted from the atmosphere back to the surface, further increasing the surface temperature $\tau_S$. This is a positive feedback loop called *water vapour feedback*. This positive feedback is manifested in Figure 3b in the additional increase in $T_S$ on the warm branch, beyond that due to the ice-albedo bifurcation. Compare this with Figure 2, where no water vapour is present. This rapid nonlinear change is due to the relatively large size of the greenhouse constant $G_{W1} = \frac{L_v}{R_W T_R} = 17.89$, in the exponent of the Clausius-Clapeyron Equation (27), as it reappears in Equation (34).

10 ### 2.3.4 Combined CO$_2$ and H$_2$O Greenhouse Gases

The combined effect of two greenhouse gases is determined by the Beer-Lambert Law as shown in Section 2.2. If $\eta_C$ is the absorptivity of CO$_2$ as in (21) and $\eta_W$ is the absorptivity of H$_2$O as in (28), then the combined absorptivity $\eta$ of these two is obtained by adding the two corresponding optical depths $\lambda_C$ and $\lambda_W$. The overall absorptivity of the atmosphere including

these two greenhouse gases and the (assumed) constant effect of clouds is, by (17),

$$\eta(\tau_S) = 1 - (1 - \eta_C)(1 - \eta_W)(1 - \eta_{Cl}) = 1 - \exp\left[-\lambda_C - \lambda_W(\tau_S)\right](1 - \eta_{Cl})$$

$$= 1 - \exp\left[-\mu \cdot G_C - \delta\, G_{W2} \int\limits_{\tau_S - \gamma Z}^{\tau_S} \frac{1}{\tau} \exp\left(G_{W1}\left[\frac{\tau - 1}{\tau}\right]\right) d\tau\right](1 - \eta_{Cl}). \tag{35}$$

The full nondimensional two-layer EBM is therefore specified by Equations (9)–(12) and (35), and the parameter values in Table 1.

## 2.4  Positive Feedback Mechanisms

The above analysis shows that there are two highly nonlinear positive feedback mechanisms in this EBM. Both serve to amplify an increase (or decrease) in surface temperature $T_S$ near the freezing point, as follows. Consider the case of rising temperature. The first feedback is *ice-albedo feedback*, due to a change in albedo in the surface equilibrium equation. As illustrated in Figure A1, if the surface temperature increases slowly through the freezing point, causing a drop in albedo, there is a large increase in solar energy absorbed, leading to an abrupt increase in temperature. The second is *water vapour feedback* in the atmosphere equilibrium equation. As shown in Figure 3, if the surface temperature continues to increase above freezing, then the absorptivity of water vapour increases dramatically, strengthening the greenhouse effect for water vapour and further increasing the temperature. Both of these mechanisms act independently of the concentration of $CO_2$ itself in the atmosphere. However, if the concentration of $CO_2$ goes up, causing a rise in temperature, then each of these two positive feedback mechanisms can *amplify* the increase in temperature that would occur due to $CO_2$ alone.

## 3  Applications of the Energy Balance Model

The goal of this section is an exploration of the underlying causes of abrupt climate changes that have occurred on Earth in the past 100 million years, using as a tool the EBM developed in Section 2. The most dramatic climate changes occurred in the two polar regions of the Earth. The climate of the tropical region of the Earth has changed relatively little in the past 100 million years. In this Section, the EBMs for the two polar regions are applied to the Arctic Pliocene Paradox, the glaciation of Antarctica, the warm equable Cretaceous problem and the warm equable Eocene problem. For comparison, a control EBM with parameter values set to those of the Tropics shows no abrupt climate changes in the Tropical climate for the past 100 million years.

## 3.1  EBM for the Pliocene Paradox

The Arctic region of the Earth's surface has had ice cover year-round for only the past few million years. For at least 100 million years prior to about 3 Ma, the Arctic had no permanent ice cover, although there could have been seasonal snow in the winter. Recently, investigators have found plant and animal remains, in particular on the farthest northern islands of the Canadian Arctic Archipelago, which demonstrate that there was a wet temperate rainforest there for millions of years, similar

to that now present on the Pacific Northwest coast of North America (Basinger et al., 1994; Greenwood et al., 2010; Struzik, 2015; West et al., 2015; Wolfe et al., 2017). The relative humidity has been estimated at 67% (Jahren and Sternberg, 2003), and this value has been chosen for $\delta$ in the Arctic EBM of this section. The change from ice-free to ice-covered in the Arctic occurred abruptly, during the late Pliocene and early Pleistocene. This is sometimes called the Pliocene-Pleistocene Transition

(PPT 3.0–2.5 Ma) (Willeit et al., 2015; Tan et al., 2018). It has been a longstanding challenge to explain this dramatic change in the climate; however, significant progress has been made recently for the case of the Greenland ice sheet (GIS). The authors of Willeit et al. (2015) have solved the "inverse problem" for the GIS by finding a schematic $pCO_2$ concentration scenario, from an ensemble of transient simulations using an Earth system model of intermediate complexity (EMIC), that gives the best fit to data from 3.2 to 2.4 Ma, taking into account the obliquity cycle. Meanwhile, Tan et al. (2018) have used an OAGCM

asynchronously coupled with sophisticated ice sheet models, to reproduce the waxing and waning of the GIS across the PPT, obtaining good qualitative agreement with ice rafted debris (IRD) data reconstructions.

Currently, there is great interest in the mid-Pliocene climate, because it is the most recent paleoclimate that resembles the future warmer climate now predicted for the Earth. Significant progress in understanding Pliocene climate has been achieved in recent years. The Pliocene Research, Interpretation and Synoptic Mapping (PRISM) project of the US Geological Survey

has contributed to this goal (Dowsett et al., 2011, 2013, 2016), as has the Pliocene Model Intercomparison Project (PlioMIP) (Haywood et al., 2011, 2016; Zhang et al., 2013). Advances in the extraction and interpretation of proxy data have given a clearer picture of the warm Pliocene climate (Haywood et al., 2009; Salzmann et al., 2009; Ballantyne et al., 2010; Steph et al., 2010; Seki et al., 2010; Bartoli et al., 2011; De Schepper et al., 2013; Knies et al., 2014; O'Brien et al., 2014; Brierley et al., 2015; Fletcher et al., 2018). At the same time, computer models have achieved closer agreement with proxy data, see

for example Haywood et al. (2009); Dowsett et al. (2011); Zhang and Yan (2012); Sun et al. (2013); Willeit et al. (2015); Tan et al. (2017); Tan et al. (2018); Chandan and Peltier (2017, 2018), and other references therein. Using a fully coupled Atmosphere-Ocean GCM, Lunt et al. (2008) considered the following forcing factors contributing to late Pliocene glaciation: decreasing carbon dioxide concentration, closure of the Panama seaway, end of a permanent El Niño state, tectonic uplift and changing orbital parameters. They concluded that falling $CO_2$ levels were primarily responsible for the formation of the

Greenland ice-sheet in the late Pliocene. Recently, Tan et al. (2018) have strengthened this conclusion.

During the Pliocene Epoch, important forcing factors that determine climate were very similar to those of today. The Earth orbital parameters, the $CO_2$ concentration, solar radiation intensity, position of the continents, ocean currents and atmospheric circulation all had values close to the values they have today. Yet, in the early/mid Pliocene, 3.5–5 million years ago, the Arctic climate was much milder than that of today. Arctic surface temperatures were $8-19°C$ warmer than today and global sea levels

were $15-20$ m higher than today, and yet $CO_2$ levels are estimated to have been $340--400$ ppm, about the same as 20th Century values (Ballantyne et al., 2010; Csank et al., 2011; Tedford and Harington, 2003). As mentioned in the Introduction, the problem of explaining how such dramatically different climates could exist with such similar forcing parameter values has been called the *Pliocene Paradox* (Cronin, 2010; Fedorov et al., 2006, 2010; Zhang and Yan, 2012).

Another interesting fact concerning Polar glaciation is that, although both poles have transitioned abruptly from ice-free

to ice-covered, they did so at very different geological times. The climate forcing conditions of Earth are highly symmetric

between the two hemispheres, and for most of the past 200 million years (or more) the climates of the two poles have been similar. However, there was an anomalous interval of about 30 million years, from the Eocene-Oligocene Transition (EOT) 34 Ma, to the early Pliocene 4 Ma, when the Antarctic was largely ice-covered but the Arctic was largely land ice free. Because $CO_2$ disperses rapidly in the Atmosphere, its concentration $\mu$ must be the same everywhere, at any given time. Therefore we
seek a forcing factor other than $\mu$ to account for this 30 million year period of broken symmetry. One obvious difference is geography. Since the Eocene, the South Pole has been land-locked in Antarctica, while the North Pole has been in the Arctic Ocean. Therefore, our two EBMs for the North and South Poles have very different values for ocean heat transport $F_O$. We will show that this difference is sufficient to account for the gap of 30 million years between the Antarctic and Arctic glaciations.

Our Pliocene Arctic EBM brackets the Pliocene Epoch between the mid-Eocene (50Ma) and pre-industrial modern times,
and it models the effects of the slow decrease in both $CO_2$ and ocean heat transport $F_O$ in the Arctic, over this long time interval. In this Arctic model, abrupt glaciation of the Arctic is inevitable, due to the existence of a bifurcation point.

During the Eocene (56 – 34 Ma), temperatures were much higher than today, especially in the Arctic (Greenwood et al., 2010; Wolfe et al., 2017; Huber and Caballero, 2011), and $CO_2$ concentration $\mu$ also was higher than today. Estimates of Eocene $CO_2$ concentration $\mu$ vary, from 1000–1500 ppm (Pagani et al., 2005, 2006), to 490 ppm (Wolfe et al., 2017). For this
EBM, we set mid-Eocene $CO_2$ at $\mu = 1000$ (Pagani et al., 2005). Both temperature and $CO_2$ concentration have decreased steadily but not monotonically, with many fluctuations, from their Eocene values to pre-industrial modern values. The overall decrease in $CO_2$ concentration observed since the Eocene may be attributed to decreased volcanic activity, increased absorption and sequestration by vegetation and the oceans, continental erosion and other sinks.

The changes in ocean heat transport to the Arctic are more complicated and derive from many factors only summarized
here (De Schepper et al., 2015; Haug et al., 2004; Knies et al., 2014; Lunt et al., 2008; Zhang et al., 2013). There was a slow drop in global sea level, in large part due to the gradual accumulation of vast amounts of water in the form of ice and snow on Antarctica. It has been estimated that the total amount of ice today in the Antarctic is equivalent to a change in sea level of about 58 m (Fretwell et al., 2013; IPCC, 2013). This drop in sea level likely reduced the flow of warm tropical ocean water into the Arctic. Against this background, several other factors came into play, due to changing geography. In the Eocene, the North
Atlantic Ocean was not always connected to the Arctic, but the Turgai Sea existed between Europe and Asia and connected the warm Indian Ocean to the Arctic, until about 29 Ma (Briggs, 1987). By the Oligocene, the Turgai Sea had closed and the North Atlantic had opened between Greenland and Norway, forming a deep-water connection to the Arctic Ocean. The Bering Strait opened and closed. During the Pliocene the formation of the Isthmus of Panama about 3.5 Ma cut off a warm equatorial current that had existed between the Atlantic and Pacific, at least since Cretaceous times. On the Atlantic side of the Isthmus,
the sea water became warmer, and became more saline due to evaporation. The Gulf Stream carried this warm salty water to Western Europe. One might expect the Gulf Stream to transport more heat into the Arctic. However, some believe that the Gulf Stream actually contributed to glaciation in the Arctic (Haug et al., 2004; Bartoli et al., 2005), as follows. Evaporation from the Gulf Stream waters contributed to rainfall across Northern Europe and Siberia, increasing the flow of fresh water in rivers emptying into the Arctic Ocean. This reduced the salinity, and hence the density, of the Arctic Ocean waters. The Gulf Stream
waters in the North Atlantic, now cooler, and denser due to high salinity, were forced downward by the less dense Arctic

waters, which began to flow into the North Atlantic on the surface. The denser Gulf Stream waters returned southward as a deep ocean current, without having conveyed much heat to the Arctic. Meanwhile the low salinity Arctic surface water, with a higher freezing temperature, began to freeze, resulting in higher albedo and accelerating Arctic glaciation. In large measure, these changing geographical factors partially cancelled each other in their contributions to ocean heat transport to the Arctic.

In the EBM, we summarize all of the above heat transport mechanisms by specifying a slow overall decrease in ocean heat transport to the Arctic, represented by the single forcing parameter $F_O$.

Figure 4a shows graphs of the surface and atmosphere equilibrium equations for the Arctic Pliocene EBM, for varying values of $\mu$. The figure shows only one surface equilibrium curve (red), with $F_O = 50\ \mathrm{Wm^{-2}}$, because the change in $F_O$ is relatively small. The blue atmosphere equilibrium curves represent values of $CO_2$ concentration $\mu$ falling from 1100 to 200 ppm, from the
top to bottom blue curves. It is clear that there may exist up to 3 points of intersection of a given atmosphere equilibrium curve (blue) with the surface equilibrium curve (red); namely, a warm equilibrium state $\tau_s > 1$, a frozen equilibrium state $\tau_S < 1$, and a third (intermediate) solution, which is always unstable (when it exists). As $\mu$ decreases and the warm equilibrium state $\tau_s > 1$ approaches the local minimum on the red S-curve; at the same time the unstable intermediate equilibrium state moves down the middle branch of the red S-curve. When they meet, these two equilibria coalesce, then disappear, via a saddlenode
bifurcation. Beyond this saddlenode bifurcation, only one equilibrium state remains, that is the stable frozen state on the left in the Figure. Dynamical systems theory tells us that following this bifurcation, the system will transition rapidly to that frozen equilibrium state. The paleoclimate record shows that $CO_2$ concentration was trending downward for millions of years before and during the Pliocene. Therefore, Figure 4a predicts that an abrupt drop in temperature to a frozen state would be inevitable, if this trend continued far enough.

In order to explore this downward trend further, we bracket the Pliocene Epoch between the mid-Eocene Epoch (50 Ma) and the pre-industrial modern era (300 years ago), and define a surrogate time variable $\nu$ by

$$t = 50(1 - \nu) \quad \mathrm{Ma.} \tag{36}$$

As reviewed above, it is believed that ocean heat transport $F_O$ decreased modestly over this time period, mainly due to the drop in global sea level, while the $CO_2$ concentration $\mu$ decreased more significantly. Therefore, we express both $\mu$ and $F_O$ as
decreasing functions of *bifurcation parameter* $\nu$

$$\begin{aligned}
\mu &= 1000 - 730 \cdot \nu \quad \mathrm{ppm} \\
F_O &= 60 - 10 \cdot \nu \quad \mathrm{Wm^{-2}}.
\end{aligned} \tag{37}$$

Here, $\nu = 0$ corresponds to estimated mid-Eocene values of $\mu$ and $F_O$ (Pagani et al., 2005; Wolfe et al., 2017; Barron et al., 1981), while $\nu = 1$ corresponds to modern preindustrial values (IPCC, 2013). Equations (37) define $\mu$ and $F_O$ as linear functions of $\nu$. In the real world, neither $\mu$ nor $F_O$ decreased linearly. This is not an obstacle for our bifurcation analysis. What is
important is that, somewhere between $\nu = 0$ and $\nu = 1$, a bifurcation point is crossed.

Figure 4b is a bifurcation diagram, which shows the dependence of surface temperature $\tau_S$ on this bifurcation parameter $\nu$. Note that, for $\nu = 0$ (mid-Eocene values, 50 Ma) only the warm equilibrium state exists. At about $\nu = 0.116$ (44 Ma) both

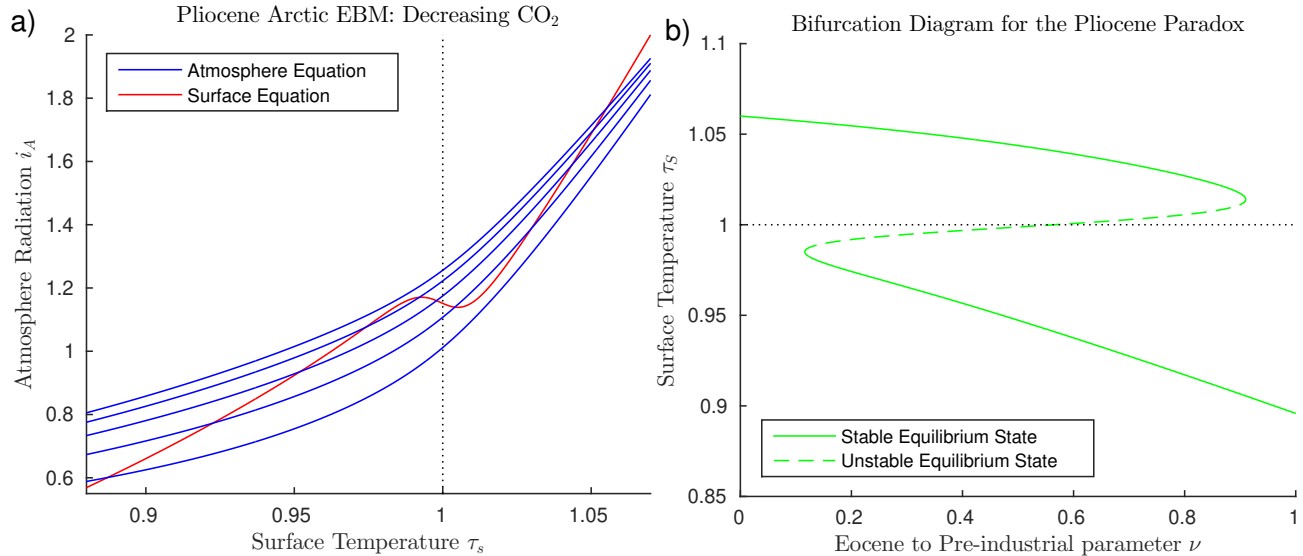

**Figure 4.** Pliocene Arctic EBM (9)–(10). Parameter values $\delta = 0.67$, $F_A = 45\ \mathrm{Wm}^{-2}$, $Q = 173.2\ \mathrm{Wm}^{-2}$, $\alpha_W = 0.08$, $Z = 9\ \mathrm{km}$. a) $CO_2$ takes values $\mu = 200, 500, 800, 1100, 1400$ ppm, from bottom to top on the blue curves, with fixed $F_O = 50\ \mathrm{Wm}^{-2}$ on the red curve. The warm equilibrium state $\tau_S > 1$ disappears as $\mu$ decreases on successively lower blue curves. b) Bifurcation Diagram for the Pliocene Paradox. Here, $CO_2$ concentration $\mu$ and ocean heat transport $F_O$ decrease simultaneously, with increasing $\nu$, $(0 \leq \nu \leq 1)$, as given by Equations (37). As $\nu$ increases, the warm equilibrium solution ($\tau_S > 1$) disappears in a saddlenode bifurcation, at approximately $\nu = 0.91$, corresponding to forcing parameters $\mu = 336$ ppm and $F_O = 50.9$. The equilibrium temperatures at this value of $\nu$ are 3.9C and -26C, and geological time about 4.5 Ma. To the right of this point, only the frozen equilibrium state exists. Between about $\nu = 0.12$ and $\nu = 0.91$, the frozen and warm equilibrium states coexist, separated by the unstable intermediate state.

warm and frozen equilibrium states exist. However, as $\nu$ increases toward $\nu = 1$, the warm equilibrium state disappears in a saddlenode (or fold) bifurcation, leaving only the frozen equilibrium state to the right of the saddlenode bifurcation. This bifurcation occurs at approximately $\nu = 0.91$, which corresponds to $F_O = 50.9\ \mathrm{Wm}^{-2}$ and $\mu = 336$, which is in good agreement with the determination (Seki et al., 2010) that the warm Pliocene pCO$_2$ was between 330 and 400 ppm, similar to today. The

5    temperatures of the warm state and frozen state, at the bifurcation value of $\nu = 0.91$ in the EBM, are $+3.9^\circ$C and $-26^\circ$C. The surrogate time of the bifurcation point, $\nu = 0.91$, corresponds to a geological time of $t = 4.5$ Ma, from (36), which is close to the time of glaciation of the Arctic in the geological record.

     The EBM plotted in Figure 4 provides a plausible explanation for the Pliocene paradox. For millions of years up to the mid-Pliocene, while the Arctic temperature remained above freezing on the warm solution branch in Figure 4b, the climate

10    change was incremental. Then the slowly-acting physical forcings of decreasing $CO_2$ concentration and decreasing ocean heat transport $F_O$ were amplified by the mechanisms of ice-albedo feedback and water vapour feedback, both of which act very strongly when the temperature crosses the freezing point of water. The EBM suggests that when the freezing temperature was

approached, the Arctic climate changed abruptly via a saddlenode bifurcation as in Figure 4b, from a warm state to a frozen state.

### 3.1.1 Permanent El Niño

Another explanation has been proposed for the Pliocene paradox. There is convincing evidence that, at the beginning of the
Pliocene, there was a permanent El Niño condition in the tropical Pacific ocean (Fedorov et al., 2006, 2010; Steph et al., 2010; Cronin, 2010; von der Heydt and Dijkstra, 2011). However, some have disputed this finding (Watanabe et al., 2011). It has been suggested that a permanent El Niño condition could explain the warm early Pliocene, and that the onset of the El Niño – La Niña Southern Oscillation (ENSO) was the cause of sudden cooling of the Arctic during the Pliocene. Today, it is known that ENSO can influence weather patterns as far away as the Arctic. However, the present authors propose that bistability and
bifurcation provide a more satisfactory explanation for the Pliocene paradox, and suggest that the concurrent change in ENSO may have been a consequence, not the cause, of the changing Pliocene Arctic climate (work in progress).

### 3.2 EBM for the Glaciation of Antarctica

Antarctica in the mid-Cretaceous Period was ice-free, and it remained so for the remainder of the Cretaceous Period and into the Paleocene and early Eocene. However, recent investigations in paleoclimate science have shown that there was an abrupt
drop in temperature and an onset of glaciation in Antarctica, at the Eocene-Oligocene Transition (EOT) about 34 Ma (Katz et al., 2008; Lear et al., 2008; Miller et al., 2008; Scher et al., 2011, 2015; Ladant et al., 2014; Ruddiman, 2014). In the mid-Cretaceous, the continent of Antarctica was in the South Pacific Ocean and the South Pole was located in open ocean, warmed by South Pacific Ocean currents (Cronin, 2010). Then, the continent of Antarctica moved poleward and began to encroach upon the South Pole towards the end of the Cretaceous, fully covering the South Pole before the end of the Eocene
(Briggs, 1987). The diminishing marine influence on the South Pole coincided with the onset of cooling of Antarctica about 45 Ma (mid-Eocene). The opening of both the Drake Passage (between South America and Antarctica) and of the Tasmanian Gateway (between Antarctica and Australia) near the end of the Eocene (34 Ma), led to the development of the Antarctic Circumpolar Current (ACC), which further isolated the South Pole from warm ocean heat transport and accelerated the cooling and glaciation of Antarctica (Cronin, 2010). Therefore, from the early Eocene to the Oligocene, the ocean heat transport to the
South Polar region decreased significantly, from near the Cretaceous value, about $100\ \mathrm{Wm}^{-2}$ (Barron et al., 1981), to a much lower value, estimated here to be $30\ \mathrm{Wm}^{-2}$. The fact that the glaciation of the Antarctic took place 30 million years before the glaciation of the Arctic very likely is due primarily to the much larger decrease in ocean heat transport that took place at the South Pole.

Because the atmosphere is well mixed, at any given time the $CO_2$ level in the Antarctic is the same as elsewhere. For this
EBM, we estimate early Eocene global $CO_2$ level as $\mu = 1100$ ppm, (Pagani et al., 2005, 2006; Cronin, 2010); decreasing to approximately modern levels by the end of the Oligocene, 23 Ma, that is to $\mu = 400$ ppm.

In the recent literature, there has been a discussion of whether the primary cause of the onset of Antarctic glaciation is the slow decline in $CO_2$ concentration, or the decrease in poleward ocean heat transport due to the opening of ocean gateways

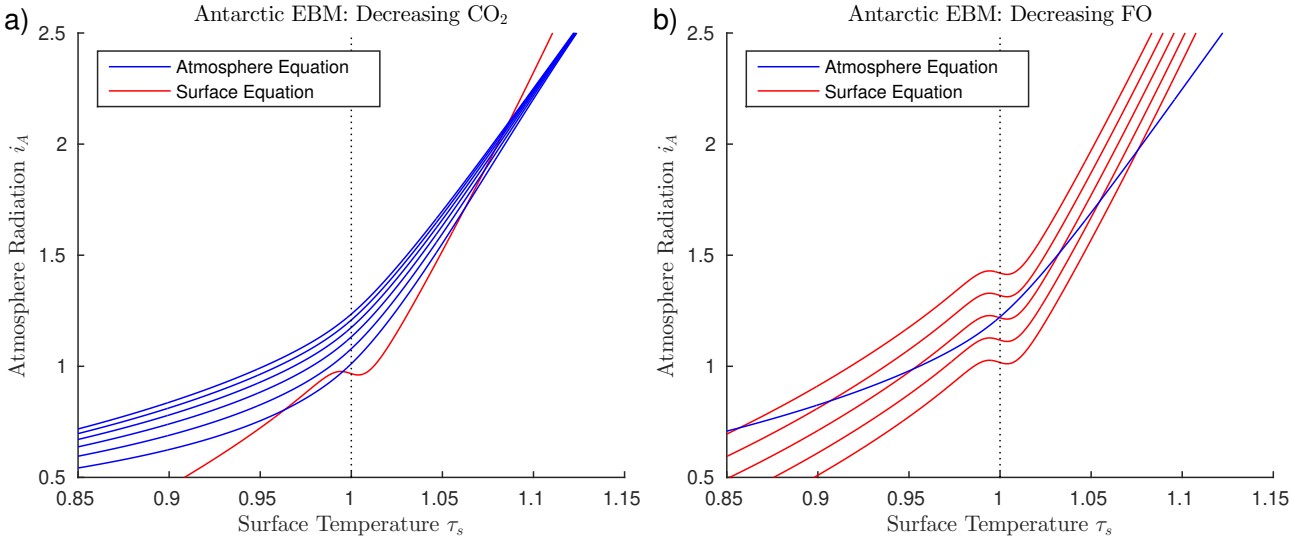

**Figure 5.** EBM for the Glaciation of Antarctica (with $\delta = 0.67$, $F_A = 45$ $\text{Wm}^{-2}$, $Q = 173.2$ $\text{Wm}^{-2}$, $\alpha_W = 0.15$, $Z = 9$ km). a) Graphs of EBM equations, with $F_O$ fixed at 90 $\text{Wm}^{-2}$ in the surface equation (red line), and with $CO_2$ concentration $\mu$ in the atmosphere equation increasing (from bottom to top blue lines) $\mu = 200, 400, 600, 800, 1000, 1200$ ppm. b) Graphs of the same EBM equations, but with $CO_2$ concentration $\mu$ fixed at 1100 ppm (blue line) and with ocean heat transport $F_O$ decreasing (from bottom to top red lines) 80, 60, 40, 20, 0 $\text{Wm}^{-2}$.

and the development of the ACC (DeConto et al., 2008; Goldner et al., 2014; Pagani et al., 2011; Scher et al., 2015). In fact, Ladant et al. (2014) state "The reasons for this greenhouse-icehouse transition have long been debated, mainly between the tectonic- oceanic hypothesis and the CO2 hypothesis". In the work Scher et al. (2015), it is observed that onset of the ACC coincided with major changes in global ocean circulation, which probably contributed to the drawdown in $CO_2$ concentration
5   in the atmosphere. Since both $CO_2$ concentration and ocean heat transport $F_O$ are explicit parameters in the EBM presented here, they can be varied independently in the model to investigate which one played a primary role.

     Figure 5 shows the solutions of the EBM equations for Antarctic parameter values, including $\delta = 0.67$, $F_A = 45$ $\text{Wm}^{-2}$, $Q = 173.2$ $\text{Wm}^{-2}$, $\alpha_W = 0.15$, and $Z = 9$ km. Points of intersection of one red line (surface equation) and one blue line (atmosphere equation) are the equilibrium solutions of the system. As noted earlier, there can be up to three equilibrium
10  solutions. In Figure 5a the ocean heat transport $F_O$ is held fixed, while $CO_2$ concentration $\mu$ is decreased, from the top to bottom blue lines. In Figure 5b the reverse is true; the $CO_2$ concentration $\mu$ is held fixed while the ocean heat transport $F_O$ is decreased, from bottom to top red lines. For a sufficiently small value of $F_O$, the warm solution disappears in Figure 5b. This occurs when the warm solution meets the unstable intermediate solution in a saddlenode bifurcation. Beyond this bifurcation, only the frozen solution ($\tau_S < 1$) exists. In contrast, reducing $\mu$ in Figure 5a does not affect the existence of the warm equilibrium, only the
15  existence of the cold equilibrium.

Figure 6 displays bifurcation diagrams, in which the surface temperature $\tau_S$, which has been determined as a solution of the EBM, is plotted while parameters in the EBM are varied smoothly, from the early Eocene values, 55 million years ago, to the late Oligocene values, 23 million years ago. These dates are chosen to bracket the time of the glaciation of Antarctica. First we introduce a *bifurcation parameter $\nu$*, which acts as a surrogate time variable. The bifurcation parameter $\nu$ is related to geological time by

$$t = 55 - 32 \cdot \nu \quad \text{Ma}. \tag{38}$$

Thus, $\nu = 0$ corresponds to the early Eocene 55 Ma and $\nu = 1$ corresponds to the late Oligocene 23 Ma. During this time, it is known that both $CO_2$ concentration $\mu$ and ocean heat transport $F_O$ were decreasing. To study the effects of the simultaneous reduction of $\mu$ and $F_O$, we express both as simple linear functions of the bifurcation parameter $\nu$, as follows:

$$\mu = 1100 - 700 \cdot \nu,$$
$$F_O = 100 - 70 \cdot \nu, \qquad 0 \leq \nu \leq 1. \tag{39}$$

Here, $\nu = 0$ corresponds to the high early Eocene values of $\mu = 1100$ ppm and $F_O = 100$ Wm$^{-2}$, while $\nu = 1$ corresponds to the low Oligocene-Miocene boundary values of $\mu = 400$ ppm and $F_O = 30$ Wm$^{-2}$. Thus, in Equations (39), as $\nu$ increases from 0 to 1, the climate forcing factors $\mu$ and $F_O$ decrease linearly from their Eocene values to late Oligocene values. Strictly speaking, these forcings did not change linearly in time; however, the important fact is that overall they were decreasing, For our bifurcation results the decrease does not need to be linear in the EBM. The atmospheric heat transport parameter $F_A$ is held constant at $F_A = 45$ Wm$^{-2}$. The solar radiation is $Q = 173.2$ Wm$^{-2}$, the warm albedo value is $\alpha_W = 0.15$, and the tropopause height is $Z = 9$ km. Other parameters are as in Table 1.

Figure 6a shows a saddlenode bifurcation at $\nu = 0.779$, corresponding to forcing parameter values $\mu = 555$ ppm and $F_O = 45.5$ Wm$^{-2}$ in the model, and to geological time about 30 Ma, assuming the linear time relation in (38). This corresponds closely to best estimates of the timing of glaciation in the geological record of about 34 Ma at the EOT. This quantitative agreement should not be taken very seriously, as it is largely due to assumptions made in the modelling. However, the *existence* of the bifurcation, implying an abrupt glaciation of Antarctica, can be taken seriously, as this is a robust (structurally stable) property of the model. The warm state and frozen state temperatures coexisting at the bifurcation point are $+3.5°C$ and $-22.1°C$. As $\nu$ increases past the bifurcation point $\nu = 0.779$, the warm climate state ceases to exist, and the climate system transitions ("falls") rapidly, from the saddlenode point to the frozen state. Thus the EBM reproduces the abrupt transition to the glaciation of Antarctica, which is seen in the geological record at the EOT.

Figure 6b explores the relative importance of decreasing $CO_2$ concentration $\mu$ and decreasing ocean heat transport $F_O$ in the glaciation of Antarctica, a subject that has been much debated in the literature (DeConto et al., 2008; Goldner et al., 2014; Ladant et al., 2014; Miller et al., 2008; Pagani et al., 2011; Scher et al., 2011, 2015). The green curve represents a scenario in which $\mu$ decreases as in Equation (39) but $F_O$ is held fixed at its Eocene value, and the magenta curve represents a case in which $\mu$ is held fixed at its Eocene value while $F_O$ decreases according to (39). In neither case does a glaciation event occur. The analysis of this paper implies that significant decreases in *both* $CO_2$ concentration $\mu$ *and* ocean heat transport $F_O$ are required to achieve a saddlenode bifurcation, reproducing the observed transition to a frozen Antarctic state at the EOT.

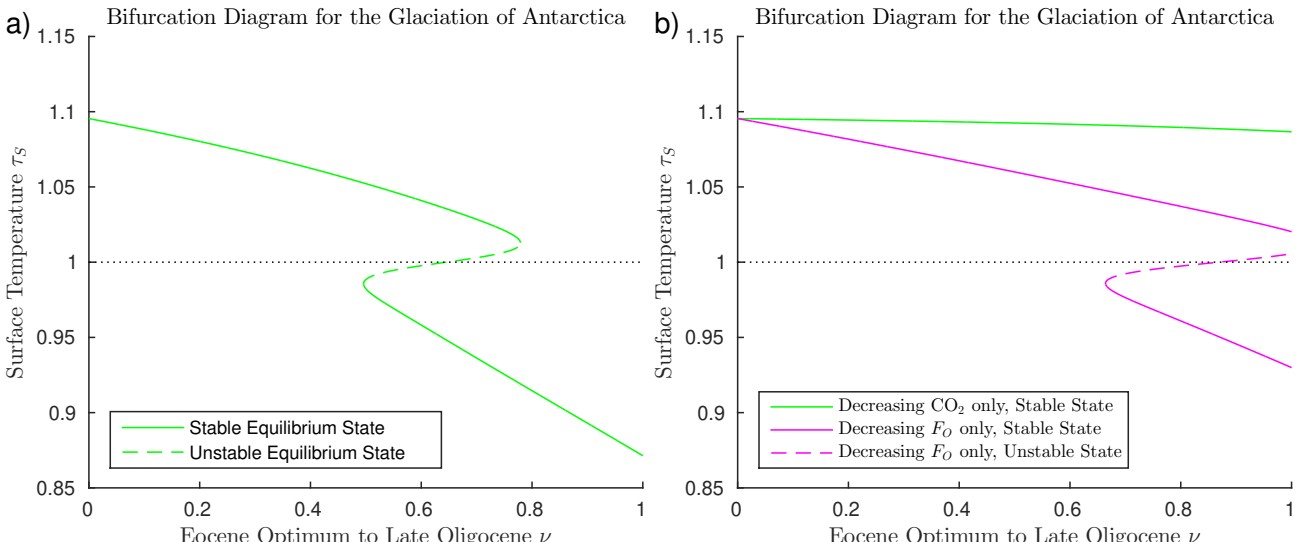

**Figure 6.** Bifurcation Diagrams for the Glaciation of Antarctica (with $\delta = 0.67$, $F_A = 45$ Wm$^{-2}$, $Q = 173.2$ Wm$^{-2}$, $\alpha_W = 0.15$, $Z = 9$ km). a) This shows a saddlenode bifurcation at $\nu = 0.779$, corresponding to forcing parameter values $\mu = 555$ ppm and $F_O = 45.5$ Wm$^{-2}$ in Equations (39). As $\nu$ increases through this saddlenode bifurcation point, the climate will transition abruptly from a warm state to a frozen state. The warm state and frozen state temperatures coexisting at the bifurcation point are $+3.5°$C and $-22.1°$C. b) Two superimposed bifurcation diagrams as in a), except that, on the green curve, $F_O$ is held fixed at its Eocene values while $\mu$ deceases as in (39), and on the magenta curve, $\mu$ is held fixed at the Eocene value while $F_O$ decreases as in (39). No saddlenode bifurcation destroying the warm equilibrium occurs in either scenario.

While the glaciation of Antarctica is an accepted fact in paleoclimate science (Miller et al., 2008; Lear et al., 2008; Pagani et al., 2011; Scher et al., 2011; Goldner et al., 2014; Scher et al., 2015) the suddenness of the climate change that occurred in Antarctica near the Eocene-Oligocene Transition (34 Ma), a time when the forcing parameters were changing slowly, has been difficult to explain. The bifurcation analysis presented here presents a simple but plausible explanation for the abruptness of this event. Furthermore, this EBM supports the hypothesis that both falling $CO_2$ concentration $\mu$ and decreasing ocean heat transport $F_O$ (due to gateway openings and development of the ACC) are essential to an explanation of the sudden glaciation of Antarctica at the EOT.

### 3.3 EBM for the Tropics

In the Tropics, many of the values of the forcing parameters are different from their values in the Arctic and Antarctic, see Table 2. The geological record shows little change in the tropical climate over the past 100 million years, other than minor cooling. Even when the Arctic climate changed dramatically in the late Pliocene, the Tropical climate changed very little.

| Forcing Parameter Values for the Tropics EBM | | |
|---|---|---|
| **Parameter** | **Modern Arctic Value** | **Tropics Value** |
| Relative humidity $\delta$ | 0.67 | 0.85 |
| Ocean heat transport $F_O$ | 20–60 $\mathrm{Wm}^{-2}$ | -50 $\mathrm{Wm}^{-2}$ |
| Atmospheric heat transport $F_A$ | 45 $\mathrm{Wm}^{-2}$ | -25 $\mathrm{Wm}^{-2}$ |
| Warm surface albedo $\alpha_W$ | 0.08 | 0.08 |
| Incident solar radiation $Q$ | 173.2 $\mathrm{Wm}^2$ | 418.8 $\mathrm{Wm}^{-2}$ |
| Tropopause height Z | 9 km | 17 km |

**Table 2.** Summary of parameters used in the Tropics EBM. Relative humidity $\delta$ is higher in the Tropics that the Arctic, and the forcings $F_O$ and $F_A$ are negative instead of positive. The insolation $Q$ is as determined by McGehee and Lehman (2012). Tropopause height $Z$ is from Kishore et al. (2006). Parameters not listed here are as in Table 1.

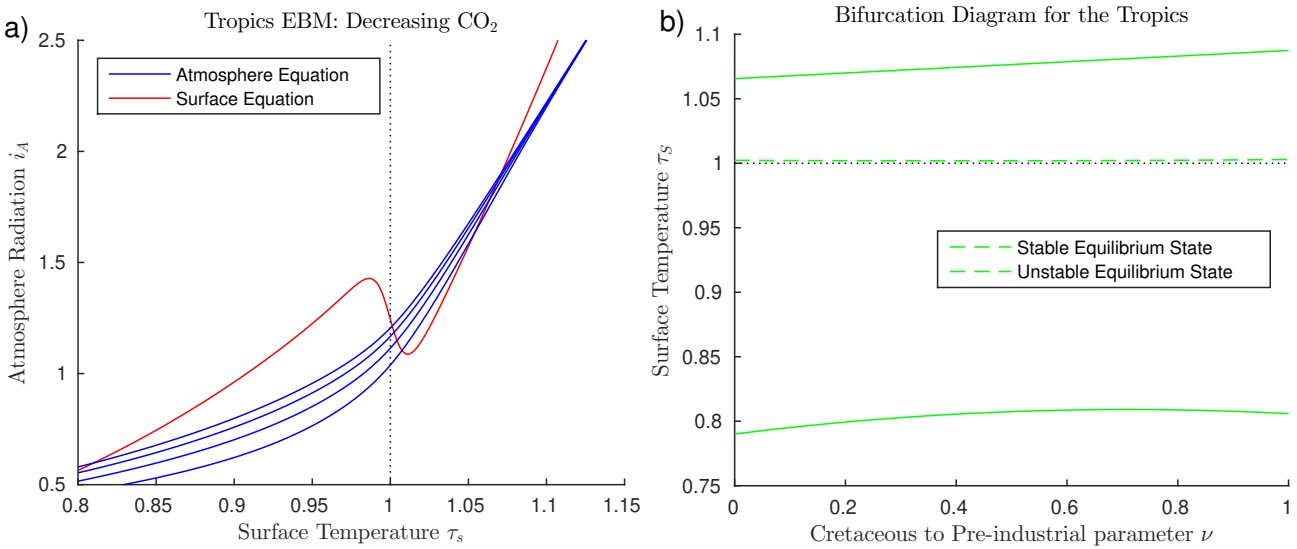

**Figure 7.** Tropics EBM, using Equations (9)–(10) and parameter values as in Table 2. a) The blue curves represent the atmosphere equilibrium equation for $\mu = 300, 600, 900, 1200$, from bottom to top. b) Bifurcation Diagram for the Tropics EBM. Here, $CO_2$ concentration $\mu$ and ocean heat transport $F_O$ are both decreasing, with increasing $\nu$, $(0 \leq \nu \leq 1)$, as given by Equations (40). As $\nu$ varies, both the warm equilibrium state ($\tau_S > 1$) and the frozen equilibrium state ($\tau_S < 1$) persist; there are no saddlenode bifurcations.

Figure 7a shows solutions of the EBM equations for Tropical parameter values. Note that at no value of the forcings used here does the warm equilibrium point approach a saddlenode bifurcation point. Thus, our EBM is in agreement with the geological record.

A bifurcation diagram is constructed for the Tropics EBM, spanning mid-Cretaceous to modern pre-industrial times, similar

5    to those for the Antarctic and Arctic glaciations. Here $\nu = 0$ corresponds to mid-Cretaceous values and $\nu = 1$ corresponds to

modern pre-industrial values. We let $\mu$ and $F_O$ decrease linearly with $\nu$.

$$\mu = 1130 - 860 \cdot \nu,$$
$$F_O = -55 + 29 \cdot \nu, \qquad 0 \leq \nu \leq 1. \tag{40}$$

The mid-Cretaceous $CO_2$ concentration $\mu = 1130$ ppm is as determined by Fletcher et al. (2008), see Table 3. The ocean heat transport $F_O = -55 \ \mathrm{Wm}^{-2}$, from the Tropics in the mid-Cretaceous, is from Barron et al. (1981). Astrophysicists have
determined that solar luminosity is slowly increasing with time (Sagan and Mullen, 1972). For the mid-Cretaceous, insolation was approximately 1% less than it is today (Barron, 1983). This difference is considered too small to be significant in our model. The bifurcation diagram for the Tropics EBM is shown in Figure 7b. Note that no bifurcations occur for parameter values relevant to the Tropics. This is in agreement with paleoclimate records that show little change in Tropical climate, even when polar climates changed dramatically.

### 3.4 EBM for the Cretaceous and Eocene Warm Equable Climate Problems

One hundred million years ago, in the mid-Cretaceous period, the climate of Earth was much more *equable* than today. "More equable" means that the pole-to-equator temperature gradient was much smaller, and also the seasonal summer/winter temperature variations were much smaller. The climate in the Tropics was only slightly warmer than today, but the climate at both poles was much warmer than today. An abundance of plant and animal life thrived under these conditions, from the equator
to both poles, including of course dinosaurs. The question of how this globally ice-free climate could have been maintained was explored in pioneering work of Barron and coworkers (Barron, 1983; Barron et al., 1981, 1995; Sloan and Barron, 1992). Barron called this the *warm, equable Cretaceous climate problem*. Early General Circulation Models (GCM) of the 1980's, adjusted to mid-Cretaceous forcing parameter values, had difficulty giving good agreement with climate proxies. In order to obtain polar temperatures in agreement with mid-Cretaceous values, these simulations typically assumed increased $CO_2$ levels
more than 4 times modern levels; but then the tropical temperatures predicted by the models were too high. Today, the issue of reproducing an equable equator to pole temperature gradient remains a challenge for modellers.

There have been many investigations of the correlation between early climate models and proxy data for the Cretaceous climate (Pagani et al., 2005; Bice et al., 2006; Donnnadieu et al., 2006; Cronin, 2010; Ruddiman, 2014). Recent studies have succeeded in narrowing the gap (Craggs et al., 2012; Bowman et al., 2014; O'Brien et al., 2017; Lunt et al., 2016; Ladant
and Donnadieu, 2016). Based on ocean drilling samples, Bice et al. (2006) estimate Cretaceous $CO_2$ concentrations between 600 ppmv and 2400 ppmv, and tropical Atlantic upper ocean temperatures between $33°C$ and $42°C$. Based on fossil samples, Fletcher et al. (2008) estimate mid-Cretaceous atmospheric $CO_2$ concentrations of 1130 ppmv and we choose their estimate for use in this EBM, see Table 3.

Paleoclimate data from the mid-Cretaceous show little difference in climate between the two warm poles at that time (Barron
et al., 1981; Barron, 1983). The major difference in forcings between the two poles at that time was that ocean heat transport $F_O$ was much higher in the Antarctic than the Arctic, as shown in Table 3. This is due to the location of the South Pole in open ocean during the Cretaceous, as the continent of Antarctica had not yet drifted to its present position over the South Pole.

| Forcing Parameters for the mid-Cretaceous EBM, 100 Ma | | | |
|---|---|---|---|
| Parameter | Arctic Value | Antarctic Value | Source |
| $CO_2$ level $\mu$ | 1130 ppm | 1130 ppm | Fletcher et al. (2008) |
| Relative humidity $\delta$ | 0.67 | 0.67 | Jahren and Sternberg (2003) |
| Ocean heat transport $F_O$ | 22 W m$^{-2}$ | 60 W m$^{-2}$ | Barron et al. (1981) |
| Atmospheric heat transport $F_A$ | 56 W m$^{-2}$ | 25 W m$^{-2}$ | Barron et al. (1981) |

**Table 3.** Summary of parameters used in the Polar mid-Cretaceous model. The main difference in Cretaceous climate between the two poles was that ocean heat transport was much higher in the Antarctic. Parameters not listed here are the same as in Table 1. The values of $F_A$ and $F_O$ shown here are estimated from Barron et al. (1981).

Estimates for atmospheric transport are also different (Barron et al., 1981). Therefore, we model the mid-Cretaceous climate at the two poles using Equations (9)–(10), with the values from Table 3. The EBM equilibrium curves are shown in Figure 8, where it is clear that a warm ($\tau_S > 0$) and a frozen ($\tau_S < 0$) equilibrium state exists at both poles, for the given parameter values. The Antarctic warm equilibrium state is warmer than in the Arctic, because of the higher ocean heat transport $F_O$ to the Antarctic. From Section 3.3, the Tropical region of Earth also had a warm equilibrium state under mid-Cretaceous conditions. Therefore, the EBM of this paper implies existence of a pole-to-pole, warm, equable Cretaceous climate, as is seen in the geological record.

Figure 8 also implies that a frozen equilibrium state is mathematically possible at each pole during the Cretaceous. In that case, the Tropics could remain in the warm state, thus giving the mathematical possibility of a Cretaceous climate that is *not equable*, but has warm Tropics and ice-covered poles, like today's climate. This may help explain why some computer simulations, originally designed for today's climate conditions, found a mathematically existing Cretaceous climate that resembled today's climate more than the warm, equable climate that physically existed on Earth in the Cretaceous.

The climate of the Eocene Epoch ($\approx 55$ million years ago) was the warmest of the past 65 million years, but a little cooler than the Cretaceous (Sloan and Barron, 1992; Pagani et al., 2005; Cronin, 2010; Huber and Caballero, 2011; Hutchinson et al., 2018). Both poles were ice-free and the pole to equator temperature gradient was much smaller than today. As for the mid-Cretaceous, early computer simulations, based on modern climate conditions, had difficulty in reproducing the early Eocene warm equable climate (Sloan and Barron, 1990, 1992; Sloan and Rea, 1995; Jahren and Sternberg, 2003; Huber and Caballero, 2011). This discrepancy has been called the *early Eocene warm equable climate problem*. Here we combine these two "problems" under one name, as the *warm equable Paleoclimate problem*.

The main difference in forcings between early Eocene conditions and those of the mid-Cretaceous, is that the global $CO_2$ concentration $\mu$ may have been a little lower and the ocean heat transport $F_O$ to the Antarctic was less. Referring to Figure 8, this means that the blue and green atmosphere equilibrium curves move downward slightly, and the magenta Antarctic surface equilibrium curve moves upward slightly. The orange Arctic surface equilibrium curve does not change significantly. With

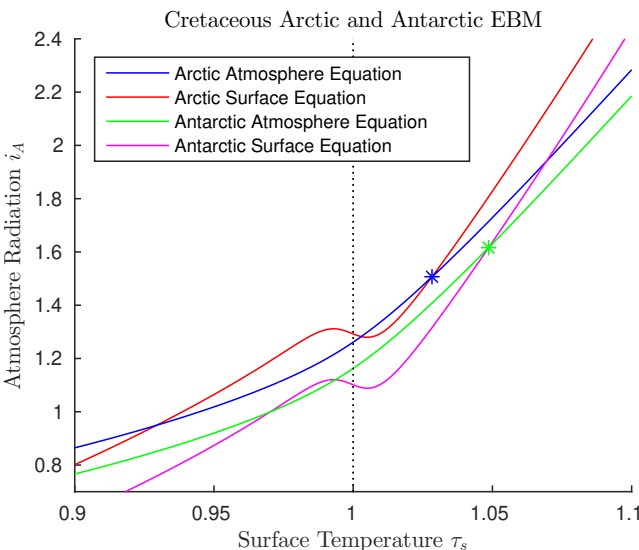

**Figure 8.** Solutions of the mid-Cretaceous EBM for both poles, using parameter values in Table 3. The blue and orange curves represent the EBM for the Arctic while the green and magenta curves represent the Antarctic. The two Z-shaped curves represent the surface equilibrium equations for the Arctic (upper orange curve) and the Antarctic (lower magenta curve) . The only differences in these two is that ocean heat transport $F_O$ is much greater in the Antarctic than the Arctic, and the atmospheric transport, $F_A$, is smaller. Both poles support both warm and frozen equilibrium paleoclimate states. The warm equilibrium states at the two poles are marked by stars.

these small changes, all of the equilibrium climate states (intersection points) persist in Figure 8. The figure for the Eocene EBM (not shown here) is topologically the same as Figure 8 for the mid-Cretaceous EBM,

Thus, the EBM predicts that a warm, ice-free equilibrium state exists (mathematically) at both poles in the mid-Cretaceous and early Eocene. From Section 3.3, the equatorial region also has a warm equilibrium state for Eocene conditions. Therefore,
5   this EBM study supports the existence of a pole-to-pole, warm equable climate in both the mid-Cretaceous and the early Eocene. Additionally, in the EBM there is co-existence of a non-equable climate, with cold poles, under exactly the same forcing conditions, which suggests the following plausible solution to the "warm equable paleoclimate problem" in both the mid-Cretaceous (Barron et al., 1981; Barron, 1983; Sloan and Barron, 1990; Barron et al., 1995) and the early Eocene (Sloan and Barron, 1990, 1992; Sloan and Rea, 1995; Jahren and Sternberg, 2003; Huber and Caballero, 2011). While the Earth's
10   climate existed in the warm equable climate state of the EBM, computer simulations of Barron and others may have correctly computed the co-existing non-equable solution.

## 4   Conclusions and Future Work

This paper presents a new energy balance model (EBM) for the climate of Earth, one that elucidates the distinctive roles of carbon dioxide and water vapour as greenhouse gases, and also the role of ice-albedo feedback, in climate change. Nonlinearity

of the EBM leads to multiple solutions of the mathematical equations and to bifurcations that represent transitions between coexisting stable equilibrium states. This EBM sheds new light on several important problems of paleoclimate science; namely, the Pliocene Paradox, the abrupt glaciation of Antarctica, and the warm equable mid-Cretaceous and early Eocene climate problems. Predictions of the EBM are in qualitative agreement with the paleoclimate record.

There has been a range of values of paleoclimate forcing parameters in the literature, in particular for mid-Cretaceous and Eocene $CO_2$ concentrations and temperatures. The specific choices made here, while informed by proxies, are somewhat arbitrary. However, this fact does not affect the validity of our main conclusions. The *coexistence of multiple solutions* and the *bifurcations* demonstrated in the EBM, are robust phenomena. That is, the existence of these multiple solutions and bifurcations will persist, over a range of values of the forcing parameters. The main conclusion of this paper, that sudden and significant

changes in climate have occurred, even while forcing parameters were changing very gradually, follows from the existence of mathematical bifurcations in the EBM, not from particular choices of the forcing parameters.

     As the paleoclimate record becomes clearer, there is growing evidence for small but rapid fluctuations in some parameter values, including $CO_2$ concentrations, over the geological time period studied here (Cronin, 2010; Pagani et al., 1999, 2005). The model of this paper assumes a smooth decline in $CO_2$ concentration. However, this does not invalidate our main conclu-

sions. The theory of Stochastic Bifurcation (Namachchivaya, 1990; Arnold et al., 1996) tells us that, if stochastic noise is added parametrically to a deterministic bifurcation problem, then typically the location of the bifurcation (in terms of the bifurcation parameter) may change, but the *existence* of the bifurcation is preserved.

     Further work on this EBM is in progress. Having demonstrated the applicability of the EBM to known paleoclimate transitions, this EBM is now being applied to anthropogenic climate change, with the goal of predicting the future climate effects

of continued increases in $CO_2$ concentration. The most significant question is whether a bifurcation is to be expected in our future. This bifurcation would lead to climate warming even stronger than that currently underway, leading possibly to a *warm equable climate*, at least in the northern hemisphere, similar to the climate on Earth before the Pliocene.

     The Equilibrium Climate Sensitivity (ECS) of the EBM, adapted to present-day satellite data, has been calculated and is presented in Appendix B of this paper.

This scalar EBM will be generalized to a two-point boundary value problem in the altitude variable $z$, using the Schwarzschild equations to replace the ICAO International Standard Atmosphere approximation for lapse rate. That change will model more accurately the behaviour of the greenhouse gases in the atmosphere. Next, that one-dimensional EBM BVP will be incorporated into a generalization of the spherical shell PDE model of Lewis and Langford (2008); Langford and Lewis (2009). We anticipate that this 3D zonally-symmetric Navier-Stokes Boussinesq model will confirm the fundamental predictions, relating

to bistability and bifurcation, of the present simple EBM. Also, it will enable the study of a third positive feedback mechanism (in addition to the two studied in this paper); namely, Hadley cell convection feedback, which influences atmospheric heat transport $F_A$.

## Appendix A:  Empirical Calibration of EBM Parameters

Some of the parameters used in the EBM have standard values, available in textbooks and reported in Table 1. Others, however, are determined empirically in this Appendix. Modern values of these parameters may be determined from today's abundant satellite and land-based data. It is assumed that the values determined here are also valid for paleoclimates.

The primary source for these empirical calibrations are the data presented by (Wild et al., 2013, Figure 4) and (Trenberth et al., 2009, Figure 3). The data from these authors are globally averaged values. We are applying our model to specific regions (the Arctic, Antarctic, or Tropics) and therefore adjust some of these values as discussed below. We also employ the model with globally averaged values to give an Equilibrium Climate Sensitivity calculation in Appendix B.

### A1   Solar Radiation

The value of the annually averaged solar radiation at either Pole is $Q = 173.2$ Wm$^{-2}$ and at the Equator is $418.8$ Wm$^{-2}$ (McGehee and Lehman, 2012; Kaper and Engler, 2013). Values from Figure 4 of Wild et al. (2013) indicate the globally averaged solar radiation is $Q = 340$ Wm$^{-2}$. Of this, $79$ Wm$^{-2}$ is directly absorbed by the atmosphere, and $76$ Wm$^{-2}$ is reflected by the atmosphere back into space (both primarily due to clouds). Hence we define the two solar radiation fractions

$$\xi_A = \frac{79}{340} = 0.2324 \qquad \text{and} \qquad \xi_R = \frac{76}{340} = 0.2235. \tag{A1}$$

Since we incorporate no information on varying cloud cover in the model, and since there is very little such information for paleoclimates, we assume these values are constant around the globe. Clearly cloud cover is correlated with surface temperature. We also tried making $\xi_A$ and $\xi_R$ vary with $T_S$ in a manner similar to how we model heat conduction/convection, $F_C$, below, but found no qualitative differences in the results. For this reason, and to keep the model simple, we have kept these values of $\xi_A$ and $\xi_R$ as global constants in our model.

### A2   Surface Albedo

The earth's albedo, $\alpha$, varies considerably depending on the surface features. Typical values are $0.6--0.9$ for snow, $0.4--0.7$ for ice, $0.2$ for crop land and $0.1$ or less for open ocean. In previous papers, including Dortmans et al. (2018), the polar albedo $\alpha$ is assumed to jump between two discrete values, a cold albedo $\alpha_C$ for the ice/snow covered surface, when below the freezing temperature, and a warm albedo $\alpha_W$ corresponding to land or open ocean above the freezing temperature; that is,

$$\alpha = \begin{cases} \alpha_c & \text{if } T_S \leq T_R, \\ \alpha_w & \text{if } T_S > T_R. \end{cases} \tag{A2}$$

This discontinuous albedo function is conceptually simple but it is not an accurate representation of what would actually happen if the polar region cooled from ice-free to ice-covered. Recall that this model represents the annually averaged climate. As the polar region cools, there will be a transition period in which warm ice-free summers get shorter and cold ice-covered winters get longer. The annually averaged albedo, therefore, would not jump abruptly from low to high constant values as in

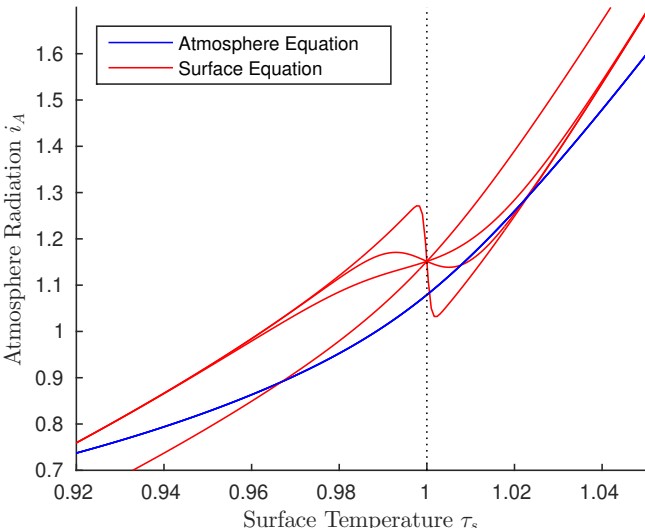

**Figure A1.** Graphs of the non-dimensionalized EBM Equations (9)–(10). The red curves represent the surface EBM Equation (10). The smooth red curves represent $\alpha(T_S)$ as in (A3), with $\omega \equiv \frac{\Omega}{T_R} = 0.001,\ 0.01,\ 0.02,\ 0.5$, on successively flatter red curves. The blue curve represents the atmosphere Equation (9). There are up to 3 intersections of the blue curve (9) with a given red curve (10) in this figure. Other parameters in the EBM are $\alpha_C = 0.7$, $\alpha_W = 0.08$, $\delta = 0.67$, $\mu = 400$, $F_A = 45\ \mathrm{Wm}^{-2}$, $F_O = 50\ \mathrm{Wm}^{-2}$, $Z = 9$ km.

(A2); it would transition more smoothly between the summer and winter extreme values. Therefore, in this paper we introduce a more realistic (and smooth) sigmoidal albedo given by the hyperbolic tangent function, Equation (5), and reproduced here:

$$\alpha = \frac{1}{2}\left( [\alpha_W + \alpha_C] + [\alpha_W - \alpha_C]\tanh\left(\frac{T_S - T_R}{\Omega}\right) \right). \tag{A3}$$

The parameter $\Omega$ determines the steepness of the transition between $\alpha_C$ and $\alpha_W$. We have chosen $\omega \equiv \Omega/T_R = 0.01$. With this
5  choice, the albedo changes by 80% (that is, it goes from 0.1 to 0.9 of the difference between $\alpha_C$ and $\alpha_W$) over a span of 6 C in annual mean temperature, which is realistic. See Figure A1.

Figure A1 shows the non-dimensional energy balance equations (9) (in blue) and (10) (in red), for a range of values of the parameter $\omega = \Omega/T_R$ in (A3). The blue (atmosphere) curve may have up to three intersections with a red (surface) curve, implying the existence of multiple equilibrium states.

10  **A3  Average Global Surface Temperature**

Both Wild et al. (2013) and Trenberth et al. (2009) indicate that the globally averaged amount of infrared radiation being emitted by the surface of the Earth is $398\ \mathrm{W\ m}^{-2}$. Since the EBM assumes the Earth is a black body, the Stefan-Boltzmann law dictates that the corresponding temperature is

$$T_S^{\mathrm{avg}} = \left(\frac{398}{\sigma}\right)^{1/4} = 289.45\ \mathrm{K}. \tag{A4}$$

This temperature is equivalent to 16.3°C, which is a little higher than the accepted average surface temperature of about 14°C. This is not surprising since the former is obtained by essentially averaging $T^4$ while the latter is from averaging actual temperatures. We use the above value for $T_S^{\text{avg}}$ in the calibrations described below, since it is consistent with the EBM.

## A4  Heat Convection/Conduction and Evapotranspiration

*Evapotranspiration* (ET) is the transport of water from the surface to the atmosphere, in the form of water vapour. It combines the effects of evaporation from the surface and transpiration by plants. Globally, the largest contributor is evaporation from the surface of the oceans and the main determining factor there is the ocean surface temperature. The ET process also transports heat from the surface to the atmosphere, in the form of latent heat. Recently Wang et al. (2010) have shown that global ET has been increasing over the past several decades.

The forcing term $F_C$ in the EBM represents transport of heat away from the surface to the atmosphere, by conduction plus convection plus change of state of water. The most important of these is the upward transport of latent heat. Surface water evaporates, taking heat from the surface. As warm moist air rises and cools, the water vapour condenses, releasing its latent heat into the surrounding atmosphere. According to both Wild et al. (2013) and Trenberth et al. (2009), the magnitude of this forcing is $F_C = 104 \text{ Wm}^{-2}$. However, this is a globally averaged value and, due to strong dependence on temperature, is not

likely valid at other than the globally averaged surface temperature $T_S^{\text{avg}}$. In the past several decades, ET has increased by $0.6 \text{ Wm}^{-2}$ per decade (Wang et al., 2010). Extrapolating back 100 years, this would be $6 \text{ Wm}^{-2}$. In the past century the average global surface temperature has risen about 1°C. Thus it is reasonable to assume the dependence of $F_C$ on the surface temperature $T_S$ should have a slope near $T_S = T_S^{\text{avg}}$ of about 6. If the dependence of $F_C$ on $T_S$ was simply linear, then, given that $F_C(T_S^{\text{avg}}) = 104$, this slope would predict that $F_C$ was negative for $T_S < T_S^{\text{avg}} - 17.3 \approx -3°C$. Since negative values for

$F_C$ are unreasonable, and since there likely is still some heat transport even near freezing, instead of a linear function, we have modelled the dependence of $F_C$ on temperature as an hyperbolic function that is nearly zero for temperatures below freezing, and increases roughly linearly for temperatures above freezing. Specifically, we use the function

$$(T_S - T_R) = \frac{1}{2A_1}\left(F_C - \frac{A_2^2}{F_C}\right),$$

or equivalently,

$$F_C(T_S) = A_1(T_S - T_R) + \sqrt{A_1^2(T_S - T_R)^2 + A_2^2}, \tag{A5}$$

where $A_1$ and $A_2$ are constants. In terms of the non-dimensional variables and parameters (8), this equation becomes

$$f_C(\tau_S) = a_1(\tau_S - 1) + \sqrt{a_1^2(\tau_S - 1)^2 + a_2^2}. \tag{A6}$$

The constants $A_1$ and $A_2$ are chosen so that the forcing at the global average surface temperature $T_S^{\text{avg}}$ is 104, and so that the forcing at the freezing point is twenty percent of this value, that is, $F_C(T_R) = 0.2F_C(T_S^{\text{avg}})$. Thus

$$A_2 = 0.2 \cdot 104 = 20.8, \quad \text{and} \quad A_1 = \frac{1}{2(T_S^{\text{avg}} - T_R)}\left(104 - \frac{A_2^2}{104}\right) = 3.063, \tag{A7}$$

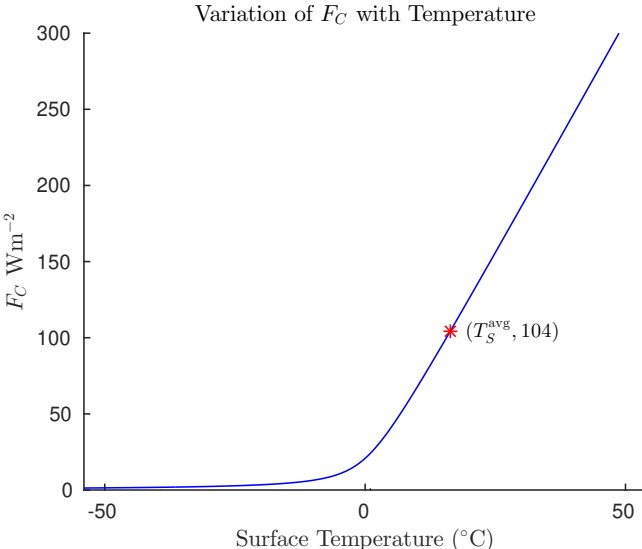

**Figure A2.** Variation of evapotranspiration/convection with surface temperature. The hyperbolic function is tangent to the horizontal as the temperature decreases, and grows almost linearly with a slope of about 6 as temperature increases.

or in terms of the nondimensional parameters,

$$a_1 = \frac{A_1}{\sigma T_R^3} = 2.650, \qquad \text{and} \qquad a_2 = \frac{A_2}{\sigma T_R^4} = 6.590 \times 10^{-2}. \tag{A8}$$

Figure A2 shows a plot of Equation (A5) for these parameter values. With these choices of the constants, the slope at $T_S^{\text{avg}}$ is 5.9, agreeing well the above argument that it should be about 6.

## 5  A5  Atmospheric Absorption of Infrared Radiation

Two of the empirical parameters in the EBM are $k_W$ and $k_C$, defined in Subsection 2.2 as the grey gas absorption coefficients for water vapour and carbon dioxide, respectively. More precisely, the absorption coefficient $k$ of a gas is its proportionality coefficient in the Beer-Lambert Law, as defined in Equation (13) of Subsection 2.2. In the *grey gas approximation*, it is assumed that the absorption coefficient $k$ is constant for all frequencies of incident radiation, and depends only on the total intensity
10  of that radiation. In reality though, $k$ is not an absolute physical constant, because gases absorb radiation at discrete spectral lines. The total amount of energy absorbed depends on the distribution of energy in the radiation at each of these spectral wavelengths. Values of $k_W$ and $k_C$ for the particular spectrum of incident radiation experienced by the atmosphere may be obtained empirically from modern era satellite and land-based measurements. We assume that $k_W$ and $k_C$ for the atmosphere do not change with time, even over the geological timescales considered here.
15    Atmospheric absorption, $\eta$, of infrared radiation emitted by the earth is, in this EBM, attributed to the two primary greenhouse gases: water vapour and carbon dioxide, and to the liquid and solid water making up clouds. According to Trenberth et al. (2009), at present as a global average, the earth emits $398 \text{ Wm}^{-2}$ of which $40.1 \text{ Wm}^{-2}$ passes through the atmosphere

to space. Thus approximately 90 percent of the radiation is absorbed in the atmosphere, that is $\eta = 0.9$. The present-day absorption factor $\eta$ can be attributed as follows (Schmidt et al., 2010): Water vapour 50%, Clouds 25% and $CO_2$ (plus other gases) 25%. Furthermore, they determined that these ratios remain unchanged, even after a doubling of $CO_2$. Therefore, setting $x = \eta_C = \eta_{Cl}$ and $\eta_W = 2x$ in Equation (17) we get

$$0.9 = 1 - (1 - x)(1 - 2x)(1 - x) = 2x^3 - 5x^2 + 4x.$$

This cubic has one real root, $\hat{x}$,

$$\hat{x} \approx 0.3729. \tag{A9}$$

In this model, since we have no data on cloud cover for paleoclimates, we make the assumption that absorption due to clouds is a constant globally and temporally, and set

$$\eta_{Cl} = \hat{x}. \tag{A10}$$

We also tried varying $\eta_{Cl}$ with temperature using an hyperbolic function like that used for $F_C$ in (A5), however we found no qualitative change in the results and therefore, for simplicity have left $\eta_{Cl}$ as a constant. To calibrate the absorption coefficient $k_C$ for the greenhouse gas carbon dioxide, we set $\eta_C = \hat{x}$ in Equation (21) yielding

$$\hat{x} = 1 - e^{-\mu \cdot 1.52 \times 10^{-6} k_C P_A / g} \quad \Longrightarrow \quad k_C = \frac{g \ln(1 - \hat{x})}{\mu (1.52 \times 10^{-6}) P_A}.$$

Using the present day value of $\mu = 400$ ppm, the value of $\hat{x}$ from (A9), the standard atmospheric pressure value of $P_A = 101.3 \times 10^3$ Pa, and the acceleration due to gravity $g = 9.8$ m s$^{-2}$ we get

$$k_C = \frac{(9.8) \ln(1 - \hat{x})}{(400)(1.52 \times 10^{-6})(101.3 \times 10^3)} = 0.07424 \text{ m}^2 \text{ kg}^{-1}. \tag{A11}$$

With this value for $k_C$, the coefficient $G_C$, defined in (20) has the value

$$G_C = 1.52 \times 10^{-6} k_C \frac{P_A}{g} = 1.166 \times 10^{-3}. \tag{A12}$$

To calibrate $k_W$ we proceed as follows. The latent heat of vaporization of water is $L_v = 2.2558 \times 10^6$ m$^2$ s$^{-2}$ and the ideal gas constant specific to water vapour is $R_W = 461.5$ m$^2$ s$^{-2}$ K$^{-1}$. Thus from (33) we have

$$G_{W1} = \frac{L_v}{R_W T_R} = 17.89. \tag{A13}$$

Dai (2006) indicates that an average value for relative humidity, $\delta$, from 1976 to 2004 for the region of the earth between $60°$S and $75°$N is 0.74. Over the same latitudes, Dai reports that averages over water and over land are 0.79 and 0.65, respectively,

except that over deserts the humidity drops to 30–50%. The polar regions are typically drier, however, there is a significant amount of water north of $75°$N and south of $60°$S, hence we have chosen $\delta^{\text{avg}} = 0.74$ as the average global relative humidity for purposes of calibrating $k_W$. In Equation (34) we set $\eta_W = 2\hat{x}$ with $\hat{x}$ given by (A9), $\tau_S$ to the present normalized global

average temperature $T_S^{\text{avg}}/T_R$, $\delta$ to the present average global relative humidity $\delta^{\text{avg}}$, $Z$ to an average global tropopause height of $Z^{\text{avg}} = 14000$ m, and the normalized lapse rate to $\gamma = \Gamma/T_R = 2.38 \times 10^{-5}$ $m^{-1}$. Using these values and inverting (34) to isolate $k_W$ we get

$$k_W = -\ln(1-2\hat{x})\gamma \left[ \delta \frac{P_W^{\text{sat}}(T_R)}{R_W T_R} \int_{T_S^{\text{avg}}-\gamma Z^{\text{avg}}}^{T_S^{\text{avg}}} \frac{1}{\tau} \exp\left( G_{W1}\left[\frac{\tau-1}{\tau}\right]\right) d\tau \right]^{-1} = 0.05905 \text{ m}^2 \text{ kg}^{-1}. \tag{A14}$$

With this value for $k_W$, the second greenhouse gas parameter for water vapour is

$$G_{W2} \equiv \frac{k_W P_W^{sat}(T_R)}{\gamma R_W T_R} = 12.05. \tag{A15}$$

## A6  Atmospheric Emission of Radiation

In previous slab models, including Dortmans et al. (2018) and Payne et al. (2015), it was assumed that the upward and downward radiation intensities from the atmosphere are equal, that is, $\beta = \frac{1}{2}$ in Equation 2. This would be the case for an actual uniform slab; however, the real atmosphere is not uniform in temperature nor density, and in fact both the temperature and density of the atmosphere are much higher in value near the surface than near the tropopause. The net effect of this non-uniformity is that, of the total radiation intensity $I_A$ emitted by the atmosphere, almost two-thirds goes back the surface and only a little more than one-third escapes to space, according to satellite and surface data (Trenberth et al., 2009; Wild et al., 2013). From Trenberth et al. (2009), the atmosphere and clouds emit $169.9+29.9$ $\text{Wm}^{-2}$ upward to space and emit $340.3$ $\text{Wm}^{-2}$ downward to the earth. Therefore, in order to have the model in this paper represent the atmosphere more realistically, we set

$$\beta = \frac{340.3}{340.3 + 169.9 + 29.9} \approx 0.63. \tag{A16}$$

instead of $0.50$.

## Appendix B:  Equilibrium Climate Sensitivity

Equilibrium climate sensitivity (ECS) is a useful measure of the sensitivity of a given climate model to an increase in $CO_2$ concentration $\mu$. It is usually defined as the change $\Delta T$ in the global mean temperature $\bar{T}$, resulting from a doubling of $\mu$, starting from the accepted pre-industrial value $\mu = 270$ ppm (IPCC, 2013; Forster, 2016; Knutti et al., 2017; Proistosescu and Huybers, 2017). The ECS provides a first-order estimate of the amount of present-day global warming predicted by a given model, as $\mu$ increases. Because ECS is a single number, it facilitates comparisons between models. It has been used extensively in the Assessment Reports of the IPCC (IPCC, 2013), where $\Delta T$ values in the range 1.5 to 4.5 C have been documented. Some have cited this wide range of ECS values as a sign of weakness of the IPCC methodology. Recently these results have been reconciled (Proistosescu and Huybers, 2017). They showed that a linear statistical analysis of historical data gives low estimates in the 1.5 to 3 C range, while nonlinear models give higher estimates.

In terms of our EBM, to apply it to a global average we make the following settings. The global mean insolation at the top of the atmosphere is $Q = 340\,\text{Wm}^{-2}$ (Wild et al., 2013; Trenberth et al., 2009). They indicate that $161\text{W}\,\text{m}^{-2}$ of sunlight is

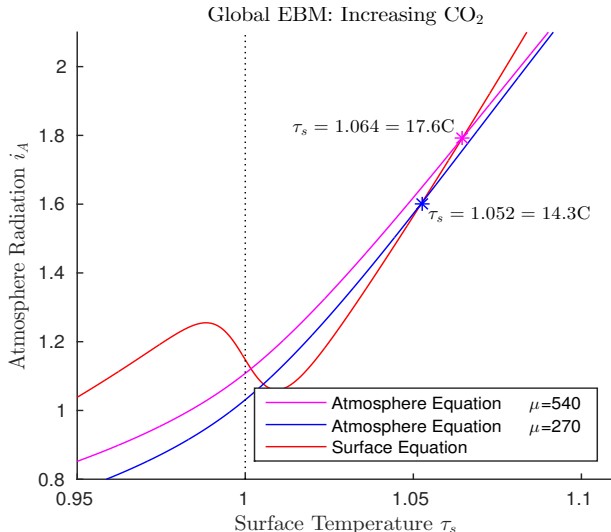

**Figure B1.** Equilibrium Climate Sensitivity (ECS) for Millennial data. The equilibrium points marked with stars show the change in $T_S$ for a doubling of $\mu$, from 270 to 540 ppm; $T_S(270) = 14.3°C$, $T_S(540) = 17.6°C$, $\Delta T = 3.3°C$.

absorbed by the earth while $24 \text{ W m}^{-2}$ is reflected back on average. Thus we take

$$\alpha_W = \frac{24}{161 + 24} = 0.13.$$

For the tropopause height $Z$ we take an average value of $Z^{\text{avg}} = 14$ km. For the global model, the ocean and atmospheric transport terms, $F_O$ and $F_A$, will be zero.

5     The ECS value we calculate will be the difference between the warm equilibrium temperatures when $\mu = 270$ and $\mu = 540$. For these calculations we took the relative humidity to be $\delta = \delta^{\text{avg}} = 0.74$, in accord with the value obtained in Dai (2006). The graphs of the surface and atmosphere equilibrium equations are shown in Figure B1. The equilibrium solutions of this global mean version of the EBM yield $T_S(270) = 14.3°C$, $T_S(540) = 17.6°C$, and $\Delta T = 3.3°C$. This is in excellent agreement with accepted values (IPCC, 2013).

10  *Code availability.*  The original Matlab code for the computations that support this research is available in the Appendix of Dortmans (2017).

*Author contributions.*  W. F. Langford contributed the original hypothesis on which the investigation is based, guided the work, and wrote much of the original paper. B. Dortmans carried out the computations required for this research and wrote an M.Sc. thesis on his work, which was the starting point for this paper. A. R. Willms co-supervised B. Dortmans' work, significantly extended his calculations, contributed many insights and guidance for this paper and wrote much of the major revision of this paper.

*Competing interests.* The authors declare that they have no conflict of interest.

*Acknowledgements.* The authors acknowledge financial support of this work by the Natural Sciences and Engineering Research Council of Canada, and thank two anonymous referees for many valuable suggestions that greatly improved the quality of the paper. The authors thank Kolja Kypke for his contributions to the final revisions of the manuscript. W. F. Langford gratefully acknowledges very helpful discussions with J. F. Basinger and D. R. Greenwood in the early stages of this work.

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
