# Peer review of "An Energy Balance Model for Paleoclimate Transitions"

_Climate of the Past, 2018_

## Referee Comment (RC1) · Anonymous Referee #1 · 11 Jul 2018

General comments:

The idea of looking at different Palaeoclimate regimes and transitions with a single EBM is an interesting approach and has good potential to increase the basic understanding of several problems that are hard to tackle with full-complexity GCMs. The authors nicely work out and motivate significant extensions to an existing EBM, mainly dividing the earth into different regions with the associated fluxes and adding the effect of water vapour as a greenhouse gas. The technical details on the model are then accompanied by an extensive overview of several experiments to test it's ability to reproduce different time frames. Especially the result showing Antarctic glaciation as a result of combined radiative forcing and meridional heat flux reduction is interesting and worth looking at in more detail.

[Figure]

A caveat here is that reduced ocean heat transport linked to a weaker ACC is probably not the sole or even main reason for reduced polar warmth; other possibilities include more subtle changes in geography, vegetation or atmospheric circulation. This also shows the still limited possibilities of the EBM that should be acknowledged; it does not include the effects of e.g. precipitation, atmospheric circulation and land/sea distribution that all play an important role in the problems visited in the paper.

While informative, especially the technical part should be shorter and more focussed and the readability/quality of the overall text improved (try to avoid repetition of similar concepts). Additionally, there is often no visual distinction between different lines in figures which occasionally makes them hard to interpret and less transparent.

Overall comparisons with previous results from both proxy and modelling studies on the palaeoclimate should be greatly improved (some recent modelling work on the Eocene, see e.g. Lunt et al. 2016,2017, Baatsen et al. 2018, Hutchinson et al. 2018 and references therein). Many claims are being made on either climatic or geological changes that are poorly referenced, some of them being invalid or uncertain. Especially the story in section 3.2 needs to be reconsidered, checked and referenced properly, while being unnecessarily lengthy in the scope of this paper.

Several nice results were obtained using the EBM, but most claims on having solved these while previous model studies using full-complexity GCMs are exaggerated or untrue.

Specific Comments:

P1

P2

L9: motivation/citation for weak seasonality in equable climate? L17: Full-complexity climate models are also highly nonlinear and have multiple states L20: geological shift of Antarctica is not that large during the Eocene, the onset and effect of a strong ACC

is highly uncertain

P3

L4: if the real world situation is more complex, there is no guarantee that similar bifurcations exist. L27: is there no effect of albedo, both atmospheric and surface? (extended later)

P4

L12: typo 'in' should be 'is' L14: two thirds reach? L15: could use some more motivation and precise citation for the b=0.63 value.

P5

In Table 1: Ts max +20C seems low for tropical, similarly 9km seems high for a polar TP.

P6

Nice to have the addition of a smooth transition between cold and warm state. Motivation for symmetric transition?

P7

L1: Motivate why I_A would be easier to observe, rather than T_A. F2: plain tanh does not add much; maybe show the full equation (5) and highlight effect of different parameters Make captions/titles/legend clearer Highlight the difference between the red lines (colour/linestyle), now they are indistinguishable Caption: rather use a) instead of 'subfigure a)' Where is figure 2b referenced in the text? L15: remove 'degree' with K

P8

L5: why use a temperature scaling that is preliminarily ill-posed?

P9

[Figure]

L11: I assume CH4 is not considered because it behaves mostly similar to CO2 and the combined effect can be considered as an increased CO2 forcing. Argument this, as in most cases CH4 is a more important GHG than CO2.

P10

L18: Addition of a second way to calculate this seems redundant

P11

F3: Although rather straightforward, it would be nice to indicate in the figure which lines belong to high/low CO2 L8: I would assume CP readers should be familiar with bifurcation diagrams and saddle node bifurcations.

P12

L3: Make this a separate sentence L10: more nearly?

P13

P14

F4: It makes little sense to have a scale up to tau_s = 1.8, meaning T_s > 400K in a climate perspective

P15

L13: no need to explain positive feedback L25: This sentence suggests that Antarctica was ice free up to 23Ma L30: double 'the'

P16

L2: would prefer the term development rather than creation, this is an extremely simplified view on a complex geological and oceanic evolution that took place over ∼20Ma L5: what motivates the lower bound of 30W/m2? L6: use Ma or million years ago, missing a reference here regarding the EOT/glaciation L7: re-phrase this; if CO2 is not the same as elsewhere, it cannot be worldwide

F5: even at 1500ppm CO2, a stable cold solution still exists in the EBM. This is a major discrepancy with higher complexity models, that generally don't allow glaciated states at >∼1000ppm and should be adressed.

P17

L17: this is a nice assumption for the experiment, but CO2 was almost certainly not linearly decreasing in this time frame (see e.g. MECO, PrOM event, post-EOT recovery). This is only mentioned later. L29: While the timing at 35Ma is nice and shows a good qualitative agreement with the geological record, the exact number is directly related to the assumptions made and not very useful.

P18

L7: reference? L9: the crossing of a CO2 threshold with/without the effect of Southern Ocean Gateways has been suggested many times before, just not in an EBM framework.

P19

L16: During the entire Cenozoic, the polar climates were in fact mostly different with Antarctica being considerably warmer than the Arctic during the Paleocene/Eocene. Going back to the mid-Cretacious, the poles were probably similar as they were both marine, but anything further back in time is highly uncertain. L18: The Arctic had no considerable land ice, but did have sea ice in the Oligocene/Miocene/Pliocene. The possible presence of land ice in the Arctic since the Eocene is also still debated. L19: In the present climate, RH is actually pretty high in most of the Arctic due to the low temperatures and marine influence. L25: Most model/proxy studies show southern high latitudes being noticeably warmer than the Arctic region. Furthermore, a high oceanic heat transport into the Arctic is unlikely considering the Arctic Ocean was mostly isolated by the geography. L29: typo: 'decreased' L31: If the glaciation of Antarctica was abrupt, then the sea level drop could not have been gradual. L34:

Model results show the exact opposite; cold, fresh waters from the Arctic flowing into the North Atlantic. L35: The North Atlantic exists since long before the Eocene, it was only narrower at not always connected to the Arctic Ocean.

P20

L1: typo: 'the' L1: the Turgai Strait probably already closed in the middle-late Eocene, missing ref here. Similarly, connections between the North Atlantic and Arctic have been there (but not continuously) since the middle Eocene. L3: The Pacific is not generally cooler than the Atlantic, locally it will be when comparing the equatorial cold tongue (EPac) and warm pool (WAtl). L7: This reasoning seems pretty far fetched and a way to get around missing the effect of enhanced precipitation in this model. In the end, qualitatively the exact same problem is being solved as for the Antarctic glaciation thus adding little to our understanding. One could instead argue that a similar reasoning can be followed to also look at the Pliocene. L15: This discussion, including F7 is mostly a copy of the one in the previous section. L30: most proxy reconstructions show little variation in $CO_2$ during the Oligocene and Miocene

P21

F7: Only a glaciated solution exists for $CO_2 < \sim 400ppm$, which disagrees with most of the Miocene/Pliocene and interglacials.

P22

L2: This is a simple solution to a simple problem: a cooling system including an ice-albedo feedback will cross a threshold for glaciation. Whether this solves the Pliocene paradox is doubtful. L5: The early Pliocene Arctic climate was relatively mild, but the climate was not equable such as that of e.g. the early Eocene. L14: The main reason for this asymmetry is simple geography, not captured in the model: Antarctica is a continent surrounded by oceans, while the Arctic is an ocean surrounded by continents. L23: This is untrue: a proper GCM spin-up starts from a specific initial condition that

is not related to the present-day climate. L29: ENSO has been shown to be present in the climate since the Eocene.

P23

L1: With the onset/strengthening of an AMOC, the equator to pole gradient should become smaller instead of larger (before glaciation). Further more, the Hadley circulation has little effect on middle-high latitude meridional heat fluxes.

P24

P25

L2: There are indeed no bifurcations, but if the change in tau_s is considered it would mean tropical temperatures decreased from ∼55C to ∼25C. A drop of 30C is certainly not small in the tropics and does not agree with the geological record. L16: More recent tropical proxies have gone up to ∼35C, while tropical temperatures in warm GCM simulations have gone down to 35-40C, bringing both estimates closer to agreement.

P26

L11: The EBM indeed shows the existence of a warm solution at the tropics and both poles, but still with a difference of 0.1-0.15 in tau_S corresponding to 27-40C which does not correspond to an equable climate (usually <20C). L14: 'The tropics' - 'their' L15: Again, this is not how computer simulations are set up to model warm climate states.

P27

L6: refrain from using the statement 'failed' here, the equable climate is a difficult problem but there has been significant modelling progress.

Similarly to 3.3, section 3.5 is a copy of 3.4 and does not add any information or physical understanding.

Conclusions:

The EBM indeed shows interesting nonlinear behaviour and is capable of showing realistic climate solutions. I only partially agree on it showing new insights in Cenozoic climate transitions or being able to better represent the cases considered.

**Zonal mean SST ($^o$C)**

| Legend |
|---|
| —— 38Ma 2x PIC |
| – – 38Ma 4x PIC |
| ⋯⋯ 38Ma Summer (4x) |
| – – 45Ma 4x $CO_2$ |
| ⋯⋯ 45Ma Summer (4x) |
| ⋯⋯ Pre-Industrial |
| ✗ 42-38 Ma Proxy |
| ◯ 38-34 Ma Proxy |

**Fig. 1.** Overview figure of SST proxies and several model studies for the middle-to-late Eocene, showing improving model-data agreement.

---

## Author Comment (AC1) · 17 Jul 2018

General comments:

The authors wish to thank Referee #1 for his/her thorough review of the manuscript and many insightful comments. All three authors are mathematicians by training and are climate science neophytes. The referee's patience in explaining more subtle climate issues is very much appreciated.

The EBM of this paper is indeed very simple and does not include the effects of precipitation, atmospheric circulation, geography or vegetation. The authors' intention was to explore the role that mathematical bifurcation theory might play in climate change, starting with the simplest possible model in order to keep the mathematics easy. This

paper is intended to be only the first in a sequence of more sophisticated models that will be closer to reality. This point will be clarified in the paper.

The results we have obtained in this paper are qualitative, not quantitative, since the model we are using does not include many contributing features such as precipitation, etc. However, we believe the model contains the most important features and the point we wish to emphasize is that the model predicts there must be a bifurcation where a warm climate state disappears leaving only a cold stable state and that this is is an important insight to understanding both the Pliocene paradox and the glaciation of Antarctica.

The authors will rewrite the text to be shorter and more focussed, avoiding repetition, and the figures will be clarified. In particular, claims made regarding paleoclimate and geological changes will be substantiated or changed. The recent publications referenced by the referee will be added. Comparisons with GCM results will be rewritten.

Specific comments:

P2 L9; L17; L20. These concerns will be addressed explicitly in the revised paper.

P3 L4: The referee is technically correct. In mathematical bifurcation theory, there is a proof that bifurcations are "structurally stable", that is, they persist in a wide class of generalizations of the model, however; it is not clear that this proof applies to something as complex as climate. L27: Albedo is treated later.

P4 L12: L14: Typos will be fixed. L15: Citation will be added.

P5 The referee has uncovered an inconsistency in Table 1. As the paper evolved, the values in Table 1 also evolved, originally reflecting Arctic values only, but later reflecting global values. Table 1 will be rewritten to be a table of Arctic values, with later sections showing corresponding Antarctic and Tropical values where appropriate.

P6 The symmetry of the tanh function is incidental here; we don't think it is important. We agree with the referee that the addition of a smooth transition between cold and

warm states is important.

P7 The referee makes a number of good points here; all will be included in the paper.

P8 L5: The authors do not understand the referee's comment that the temperature scaling is "preliminarily ill-posed". This scaling used here brings the temperature variable to neighbourhood of 1 and brings all parameters closer to 1; the most well-posed of numbers.

P9 L11: Yes; that was our idea. CH4 behaves similarly to CO2 and can be included In CO2 as a greenhouse gas. We will make this explicit; i.e. $\mu$ represents CO2 + CH4.

P10 L18: Yes, this is redundant.

P11 F3: The caption of Figure 3 a) gives the values of CO2 on the blue curves, decreasing from top to bottom. This seems clear to us. Placing that information inside the figure might clutter the figure. Also, distinguishing each of the 5 blue curves by a different colour or line type could cause clutter. Our goal was to distinguish between the surface equation (red) and each of a family of atmosphere equations (blue). L8: We will make this less pedantic.

P12 L3: OK, separate sentence. L10: We will give a range for $\delta$ and say "varies less" instead of "more nearly constant".

P14 F4: Referee is correct. Figure will be truncated.

P15 L13: The explanation of positive feedback will be abbreviated. The main point here is that there are two independent mechanisms of positive feedback in the EBM. L25: Right. We intended that this apply only to the early Paleogene, not to the entire period. We will rewrite this sentence. L30: Removed "the"

P16 L2: "Development" instead of "creation" is fine. The "extremely simplified view" described here is, we think, appropriate for our simplified model. There was indeed "a complex geological and oceanic evolution that took place over $\sim$20Ma", but most

of this is outside the scope of our model. L5: The lower value of 30 W/m^2 is just a guess, frankly, just as the upper value of 100 W/m^2 was a guess in the 1980's (by Barron). The authors had great difficulty finding reliable estimates of Ocean Heat Transport (OHT) to both poles, in different time periods. Surely, the combined effects of the ACC and the continent of Antarctica have reduced the OHT to the South Pole essentially to zero, today. If it started at 100 W/m^2 [Barron], then by the EOT it would have decreased significantly to, say, 30 W/m^2. This is plenty good enough for our topological model. L6: Ma instead of MYa. We will add references here regarding EOT / glaciation. L7: CO2 concentration is assumed the same worldwide. F5: Yes, A stable frozen solution exists in the EBM at 1500 ppm CO2, but it disappears slightly above 1500 ppm. Because this is a crude model, exact numerical values are not significant.. The important point is that the model gives a mechanism that explains the disappearance of the frozen solution, via a saddlenode bifurcation.

P17 L17: Agreed. CO2 definitely did not decrease linearly. But this is irrelevant for the bifurcation theory. All we need to know is that CO2 decreased. Then it must have crossed the bifurcation value somewhere. It is amazing good luck that it happened in the model at a time that corresponds so well to the EOT boundary. (With a little help from the authors, of course.) The MECO is an anomaly and is irrelevant to our analysis, as is the PrOM event. The Post-EOT recovery and Antarctic deglaciation ~25 Ma, and re-glaciation at ~13 Ma are fascinating, by are not explained by our analysis (yet). L29: Agreed. The referee was paying attention to notice this. But it is very satisfying for the modellers that they were able to hit the EOT boundary so accurately. We will add a disclaimer. The good numerical agreement here should not be taken too seriously.

P18 L7: References added. L9: It is true that "the crossing of a CO2 threshold with/without the effect of Southern Ocean Gateways has been suggested many times before". The relative importance of these two mechanisms is openly debated in the literature. We think our paper is the first to indicate that both mechanisms are necessary, to cross the glaciation threshold, and that the suddenness of the transition may

be explained by a bifurcation.

P19 L16: Okay, we will remove speculation about pre-Cretaceous climates. L18: Okay, we will change "the Arctic was ice-free" to "the Arctic was largely land ice free". L19: Okay, we will remove the comparison to the relative humidity of today's climate. L25: Okay, we will remove the implication that the Arctic was warmer, and also reduce the influence of ocean heat transport. L29: Typo fixed, thanks. L31: The EOT marked an abrupt initiation of glaciation in the Antarctic. However, the accumulation of ice/snow in the Antarctic was gradual, due to the low relative humidity. The depth of ice/snow in Antarctica went from meters to kilometres over a period of millions of years and sea levels dropped inversely. L34: We will note the existence of cold fresh waters flowing out of the Arctic, into the North Atlantic. L35: Point taken. The North Atlantic existed, but was only narrower and not always connected to the Arctic Ocean.

P20 L1: Typo fixed. L1: We will add a ref. for the closing of the Turgai Strait, date its closing more accurately. L3: We will clarify that the "cooler Pacific" reference is only local to this region. L7: This reasoning at first seemed far fetched to us too, but the papers of Haug et al. (2004) and Bartoli et al. (2005) convinced us. It is true that our simple model does not include the effect of enhanced precipitation directly. However, the work of Haug et al. (2004) and Bartoli et al. (2005) makes a strong argument that the net effect of the mechanisms described here was to reduce the ocean heat transport from the North Atlantic to the Arctic. This we can use in our EBM, and thus indirectly include the effects of enhanced precipitation. It is true that qualitatively the same problem is being solved for the Antarctic and Arctic glaciations. Both involve the same saddlenode bifurcation in our EBM. But, the important question remains: Why did these two glaciations occur at times separated by 30 million years, when for most purposes the Earth has a North-South symmetry? Our model gives a plausible answer to this question, by including both global $CO_2$ concentration and ocean heat transport to each pole, in the calculation. L15: This discussion will be abbreviated. L30: Pagani et al. show continued decrease in $CO_2$ levels during the Oligocene, but nearly constant

CO2 in the Miocene. This suggests that the decrease in ocean heat transport occurring at this time, as described in the paragraphs above, though small, played an important role in the final Pliocene transition.

P21 F7: Yes, in Fig. 7a), only the frozen solution exists for CO2 less than about 400ppm. However, this is true only for the particular choices of F_O and F_A indicated in the figure. Since there is great uncertainty in the values of F_O and F_A, the quantitative predictions of the EBM can not be taken literally. What is important here is that, as CO2 decreases (blue curve moving downward) and F_O decreases (red curve moving upward), a saddlenode bifurcation is inevitable, and this bifurcation corresponds to an abrupt glaciation of the Arctic. This is a qualitative, not a quantitative result. Also, the differences in conditions in the Arctic and Antarctic account for the multi-million year gap in the timing of the two glaciations. The interglacial oscillations appear to have been driven by Milankovitch cycles, a phenomenon well outside the scope of this paper.

P22 L2: As noted in the paper, there are several "Pliocene Paradoxes" to be considered here. We believe that our EBM sheds some light on the understanding of all 4 of the "paradoxes" listed here. Still, this simple model is not the final word on any of them. L5: This is a matter of degree. The early Pliocene was not as equable as the early Eocene, but was more equable than the late Pliocene. L14: Yes. The underlying cause of the asymmetry was simple geography, as the Referee points out. Exactly how this asymmetry in geography manifests itself in the asymmetry of the polar glaciations is investigated in this paper. L23: The authors are not experts on GCM spin-ups, and they thank the referee for this insight. The text will be corrected. L29: Fedorov et al (2006,2010) announced that there was a permanent El Nino condition in the early Pliocene. This is not inconsistent with ENSO conditions in the Eocene. The authors do not take responsibility for claims of a permanent El Nino in the early Pliocene,

P23 L1: This sentence refers to the equator to pole temperature gradient, post-glaciation, which clearly increased. The remainder of this paragraph, concerning

"Hadley cell feedback", is speculation at this time.

P25 L2: Yes, a drop on temperature in the Tropics of 30C, in the 100 million years from mid-Cretaceous to Pre-industrial, is excessive. However, this simple EBM is qualitative, not quantitative, and this number should not be taken too seriously. The authors had difficulty finding estimates of heat transports (F_O and F_A) away from the Tropics, so the values used here are only estimates. The important take-away from this Tropics model analysis is that (a) the average Tropical temperature did decrease, and (b) there were no bifurcations. L16: Thank you for this information.

P26 L11: Yes, the predicted equable pole-to equator temperature difference is excessive, but again the discrepancy is quantitative, not qualitative. L14: Corrected. L15: Thank you. This will be corrected.

P27 L6: We will expunge the word "failed" and its synonyms, in all reference to GCMs in this paper. Thank you for pointing us to recent progress. Section 3.5: The authors agree with the Referee that there is nothing new, either mathematically or physically, in this Section. The authors chose to separate the "warm equable mid-Cretaceous climate problem" and the "warm equable early Eocene climate problem" into different sections, because they have been presented in the literature as two independent problems. (Perhaps this is because the proxy records are separated by almost 50 million years.) From our point of view, they are the same problem. Would the Referee approve a rewriting that combined Sections 3.4 and 3.5 into one section, "The mid-Cretaceous and early Eocene warm equable climate problems"? This would remove some redundancies.

---

## Referee Comment (RC2) · Anonymous Referee #2 · 29 Jul 2018

**Review on the manuscript (cp-2018-56) entitled "An Energy Balance Model for Paleoclimate Transitions" by Brady Dortmans et al.**

This paper aims to revisit some of the major climate transitions of Mesozoic and Cenozoic using a new EBM model and to study bifurcations. Specifically the authors apply the EBM for mid-Cretaceous, early Eocene, Eocene/Oligocene (E/O) transition and Pliocene.

In a first part the authors introduce the "paradoxes" they intend to solve using new EBM. Then, in a second part, they describe the novelty of this new model and finally, in a third section, they apply this tool to different climatic transitions.

The paper is well written and tackles an interesting topic but, from my point of view, suffers from many caveats that I will describe below.

**Major comments**

1. Presentation of the climate paradoxes

Concerning the use of a large hierarchy of models to investigate different climatic transitions, I fully agree that this approach is necessary, mainly because sophisticated AOGCM can only investigate a few numbers of trajectories whereas EBM and EMIC can provide a large number of experiments and investigate a larger range of possible trajectories. This has been illustrated for instance by Roche el al., Nature, 2004 and Claussen et al., GRL, 1999.

The authors continue a long lasting history with pioneering studies of Budyko, Sellers and more recently Paillard et al., Nature, 1998, with more conceptual models, Ganopolski et al., Nature, 2001 using more complex EMIC and Stap et al., Climate of the Past, 2017 with an EBM, for instance.

My major criticism concerns the real added value of this study and the authors should more clearly pinpoint which "paradoxes" they solve with respect to the abundant literature already published on this key transition. Moreover, the bibliography concerning each period has to be updated. Since Crowley and Barron pioneering studies, in the nineties, much progress has been done with the large hierarchy of models which doesn't appear in this paper. Indeed, the transitions that the authors tackled in this paper and depicted as paradoxes or problems have been investigated by major publications which solved many problems. The main caveat with this paper is that the authors did not present these studies and gave a dated view of these questions.

In section 3, I give more details for each period including some references which are not exhaustive and the authors should really provide a better and updated list of appropriate references to clarify first if the paradox they consider is still real and what problem they solve exactly.

2. Model description, validation and sensitivity

A second caveat is that the paper is referring continuously to a recent publication (Dortmans et al., 2017) which makes the paper sometime difficult to read. Moreover, despite a detailed description on the different processes included in this model it is not always clear to understand what could be the climatic consequences of this improvement compared to previous EBMs studies.

What is crucially missing is the validation and sensitivity of the model. A rapid validation of the model for present day climate and cryosphere and sensitivity to doubling $CO_2$ scenarios or glacial-interglacial oscillations would bring some credit to the model capability before testing it for deeper time periods.

In their conclusion, surprisingly, the authors claim that after solving difficult problems for deeper time that are associated with changes in topography, $CO_2$, vegetation…they will study future climate. But in fact, we would be very interested to know what is the sensitivity of the EBM to doubling $CO_2$ in present day configuration. Indeed, there are a large bunch of model results on this transition. It would be therefore useful to compare the result of this EBM to other model results.

The fact this EBM is dividing the Earth in 3 latitudinal boxes (Arctic, Antarctica and tropics) with different properties for each box is puzzling because the climate at the end is global and it is not really possible to separate and optimize independently each box. For instance, for mid-Cretaceous (O'Brien et al., 2017, ESR and Ladant & Donnadieu., 2016, Nature Com) or even for early and mid-Pliocene, the tropical response is not completely clarified and the data model comparison for these periods are not completely robust and therefore the ocean meridional circulation from the Tropics to higher latitudes is also an open question (Z. Zhang et al., 2013, Clim Past). The authors should discuss in more details the consistency of their results over the 3 boxes and their interaction between these boxes.

For periods corresponding to ice sheet build-up (Eocene, Oligocene, Pliocene, Pleistocene) the authors should discuss the limitation of EBM to assess the correct computation of ablation, accumulation to the lack of representation of hydrologic cycle.

3. Application to different paradoxes

This part is the weakest for me due to the fact that many important references are missing and it is not always clear what in which extent this new EBM solves or clarifies these issues.

**Concerning Eocene/Oligicene transition:** the first paradox raised by the authors has been deeply explored and the bibliography the authors depicted is rather short. For instance specifically on the evolution of climate and ice sheet at the E/O transition a model study Lear et al, Geology, 2008; Scher et al., Geology, 2011; Ladant et al., Paleoceanography. 2014 see reference herein pointed out and explain many features of this glaciation for pCO2 values that are in rather good agreement with reconstructions of the literature even if there are still some uncertainties.

Antarctica tectonically achieved a polar position already circa 90 Ma ago and the ice cap is triggered only 34 Ma ago (see recent publications of Ladant, De Conto and Pollard)

I don't get the feeling the approach of the authors with their new EBM brings new insight when compared to recent publications. Moreover the exact occurrence of the transition seems more like a tuning result rather than a prediction.

**Concerning the "Pliocene" paradox**:

The conclusion summarizes (page 19) the four different paradoxes for Pliocene the authors tackle in this section. To my opinion, some are not really paradoxes and other may be irrelevant.

Paradox 1, with long lasting climate simulation at 410 ppmv for present day configurations, similar to Pliocene pCO2 reconstructions used in PLIOMIP1 and PLIOMIP2 for instance (see Haywood et al. for boundary condition description), the equilibrium achieved in several centuries would be more similar to Pliocene climate. Therefore with similar pCO2, future climate would be close to Pliocene. Therefore, this transition is more relevant of a radiative threshold on pCO2 (see recent publications Wiley et al., Tan et al. for modeming and Martinez-Botti and Seki for pCO2)

Concerning the second paradox, many studies, especially Lunt in Nature 2008, have demonstrated, using different forcing factors, that the major cause of Pliocene transition is pCO2 decrease. Such behavior has also been depicted by Deconto and Pollard.

Therefore, the others should clarify exactly what is their own contribution to solving these two paradoxes.

The decrease of pCO2 from Eocene to Pliocene will lead to the onset of Antarctic 34 Ma ago associated with tectonic and seaways but it will be necessary to wait much longer that pCO2 reach around 300 ppm for Greenland inception due to much unfavorable conditions. See Tan et al., EPSL, 2017 and also all the papers on cryosphere and climate evolution published by De Conto and Pollard. I don't really understand what is new in this third paradox

The fourth paradox is just a misunderstanding. GCM simulations did not proceed as written by the authors ". The EBM suggests that these GCM simulations, starting with today's climate and moving backward in time, would have remained on the stable frozen
25 climate state of the bifurcation diagram in Figure 7 b), and thus failed to "see" the coexisting warm state.". First, GCM do not move backward in time, and second, they are deterministic and of course cannot capture two different equilibrium modes. these models prescribed boundary conditions and a starting state. I believe that the authors mean that because some of these models start their simulations from a present day they are biased by cold conditions and only reach a cold solution and miss another equilibrium. But this point of view is over simplified. Long transient simulations during Cenozoic of De Conto and Pollard are able to reproduce these glaciations onsets.

**Concerning mid-Cretaceous climate paradox**:

Once again, since Barron's pioneering studies, many recent studies revisited the issue of cretaceous climates and pointed out a large control of paleogeography on climate evolution. Indeed, a major

reason for small temperature changes in the Tropics is the polar amplification. Moreover when dealing with deep times the continental distribution is crucial. When neither land nor ice cap is present at the pole, the equator to pole thermal gradient is indeed much flatter. Therefore, the paleogeographic configuration plays an important role for instance during mid-Cretaceous (cenomanian) with high sea-level and smaller continent areas. The authors mostly cited the pioneering studies of Barron and one more recent study (Cromin, 2010) but there are plenty of simulations which depict a more accurate view on this topic for instance Donnadieu et al. ,EPSL, 2006 and Ladant & Donnadieu, Nature Com, 2016 and references herein.

In fig. 9, the EBM simulation shows a similar result for the Arctic and Antarctic boxes which is not surprising because of symmetric forcing factors. It would be interesting to show the result for the tropical box and also to investigate an asymmetric forcing.

**Minor comments**

- In the section 2.4 "Positive Feedback Mechanisms", the authors should clarify what is really new here. As far as I know, many EBM models, as for instance those currently used by Stap et al., Climate of the Past, 2017 or Weaver et al., Journal Atmosphere-Ocean, 2001, already included these feedbacks. Maybe the computation of GHG is different but then the comparison with other EBMs should be done. Moreover, if there is some added value that doesn't exist in any EBM yet, the authors should describe its effects on climate simulation.

- In the section 3.2.1 "Permanent El Niño and Hadley cell feedback" : Indeed papers as Heather et al., GRL, 2015 suggested permanent El Niño at early Pliocene, but this is not yet a consensus view. Moreover, Lunt et al., Nature, 2008; and many other papers demonstrate that these warmer tropics do not explain the shift to perennial ice sheet over Greenland. Therefore, the authors should discuss in more details, in this section, the potential role of permanent El Niño. Concerning changes in Hadley cell during Pliocene, there are also studies comparing tropical circulation from Pliocene to present day that could be useful to the authors (Sun et al., Climate Past, 2013 and Sun et al., Climate Dynamics, 2018).

- The authors write: *"It complements, rather than replaces, more detailed General Circulation Models (GCM)."* I suggest to write instead for instance: "in complement to GCM, very sophisticated models, including a lot of 3D processes are only able to run some climate trajectories;  EBM and EMIC may explore more possibilities and investigate climate transitions (tipping points) but with major simplifications.

- The authors should be more careful concerning the ocean dynamics changes both for early Pliocene (Lawrence et al., Paleoceanography and Paleoclimatology, 2009 and De Schepper et al., Nature Com 2015) and Oligocene/Eocene (Miller et al., The Geological Society of America, 2009).

**Conclusion:**

In summary, looking to all the applications for which the authors apply this new EBM, I feel very uncomfortable because the presentation of the issues lacks of knowledge on recent studies but also does not bring new insight on these issues.

Finally, I suggest at this stage, to reject this manuscript. I believe there is room for large improvement in two major directions:

1. In demonstrating the validation of the EBM for present day and for simple sensitivity experiments as doubling $CO_2$ and glacial/interglacial transition to test the sensitivity of EBM before going to deeper time climate transitions.
2. Depicting a better and updated insight in the recent bibliography concerning each paradox to show more clearly what is the added value of these new simulations in understanding major transitions.

---

## Author Comment (AC2) · 8 Aug 2018

Manuscript CP-2018-56 AC Reply to Referee#2

The authors wish to thank Referee#2 for insightful comments on the manuscript. We are grateful for the time and effort taken by both Referees to write such detailed reviews, and also for the updated references to the paleoclimate literature. The point of view given by the references in the original manuscript is indeed dated, and this will be corrected with more current references as suggested by the Referees.

Our comments in this Reply follow the same order as the Referee#2 comments.

Major comments

[Figure]

**1. Presentation of the climate paradoxes**

The authors agree with the Referee that a large hierarchy of models to investigate different climate transitions is necessary. The papers of Roche et al. (2004), Claussen et al. (1999), Paillard et al. (1998) and Ganopolski et al. (2001) illustrate this nicely. The authors found them to be informative reading. However these four papers deal with more recent climate changes than considered in the manuscript, and therefore have not been added to the bibliography. The glacial-interglacial cycles of the Pleistocene are believed to be driven primarily by variations in Earth's orbital parameters, which are not present in the EBM of the manuscript, and thus it can not address these phenomena.

However, the authors found the paper of Stap et al. (2017) extremely interesting. Those authors found hysteresis in the relationship between $CO_2$ and temperature, over the past 38 million years, in a simple one-dimensional ice sheet model. Here "hysteresis" means the co-existence of two different mathematically-possible stable climate states. The Earth climate system can be in only one state at a given time, and the selection of which state is occupied depends on the past history of the state of the system. Stap et al. (2017) found both a warm equable state and a cold state, depending on the initial conditions chosen for their run. This result is in qualitative agreement with the predictions of our EBM. The authors have long wondered if any of the GCM studies of paleoclimates have found multiple stable climate states, for the same values of forcings. We would be very grateful if the Referee could point us to any other publications that show hysteresis or bistability in a climate model.

As for the "real added value of this study", the authors feel that their main contribution is a demonstration that a simple EBM based on sound physical principles, which are intrinsically nonlinear, can exhibit hysteresis (or bistability). That is, mathematically the EBM may have two different stable equilibrium states; one warm and equable and the other non-equable with frozen poles. The actual climate state of the Earth, or of a GCM experiment, can exist in only one of these two states. In reading the classic papers of Barron and others on the "warm equable climate problem", the authors were led

to conjecture that a solution to Barron's problem may be that the Cretaceous Earth's climate state and the solution state of Barron's early GCM experiments were simply exhibiting two different climate states, both of which were mathematically correct. The authors can not prove, from their simple EBM, that they have "solved" Barron's problem. This is only a suggestion. However, they would be very interested in exploring with GCM experts the possibility of multiple stable equilibrium states in a paleoclimate GCM.

Similarly for the "paradoxes" discussed in the manuscript, the authors can not claim to have solved them with a simple EBM calculation. Any such claims will be modified in revising the manuscript. Rather, we hope that this paper will lead to further investigations with more sophisticated models that will confirm (or refute) our conjectures. In addition to suggesting bistability, our EBM suggests that mathematical bifurcations may account for abrupt climate changes, such as occurred at the Eocene-Oligocene Transition at the South Pole, and in the Pliocene Transition at the North Pole. It has been recognized for some time that these abrupt transitions were some form of "tipping points". To a mathematician at least, it is very satisfying to think that these tipping points can be explained as mathematical bifurcations, which are unavoidable in the EBM.

A great deal of progress has been made on these problems and paradoxes in the climate science literature, since Barron's pioneering work. Thanks to the Referee's suggestions, the authors can and will cite this exciting current work, and clarify how the present work relates to current work in climate science.

2. Model description, validation and sensitivity

The Conference Proceedings publication (Dortmans et al. 2017) presents an earlier version of the EBM and a preliminary analysis of the Pliocene paradox. We will make the current manuscript more self-contained and independent of that proceedings paper.

The authors will clarify the climatic consequences of this improved EBM compared to previous EBM studies.

The authors have performed an Equilibrium Sensitivity Analysis (ESA) of the EBM under modern conditions, and plan to publish our ESA results and comparisons with other ESA results in a follow-up paper. ESA has become useful for comparisons in the IPCC reports. The main focus of this second paper will be anthropogenic climate change. We will not study glacial/interglacial oscillations because the mathematical analysis is quite different for orbital forcings. Journals in climate science tend to focus either on paleoclimates or on modern-day climate change, so we have split our work accordingly. We chose to apply our model first to paleoclimate changes, because today much is known about the paleoclimate changes addressed in this manuscript; before, during, and after the changes took place. This enabled us to "calibrate" our EBM before applying it, with greater confidence, to future climate changes which are still speculative.

It is true that changes in climate are influenced by many factors, including changes in topography, vegetation, etc. The authors' goal in this study was to find the simplest climate model that could address some well-known paleoclimate paradoxes. This model includes the effects of both $CO_2$ and $H_2O$ as greenhouse gases, in a more physically realistic way that was the case for previous EBM studies, and includes ocean and atmospheric meridional heat transport, but ignores other forcing factors. We feel that our EBM has been remarkably successful in achieving this goal.

Regarding the separation of the Earth's climate into three latitudinal models, for Arctic, Antarctic and Tropical climates, respectively, this was done to improve resolution over traditional EBM's which only consider a globally averaged energy balance. With a globally averaged EBM, one can not study Arctic amplification, for example. In the manuscript, the polar models are coupled to the tropical model indirectly, through the ocean and atmosphere transport terms, which are adjusted by hand. The authors had difficulty in finding values for these meridional transport terms; in this regard the Referee's references are useful. The paper of O'Brien et al. (2017) provides valuable detailed information about Cretaceous climates. The consistency of our results over

the three boxes will be discussed in more detail. In future work, these separate models will be coupled directly in pairs, and then later combined into a continuous meridional model.

Regarding ice-sheet build-up and ablation, these can only be estimated in this simple model. The important consequence for the EBM is the change in surface albedo; high albedo below the freezing temperature and low albedo above freezing.

3. Application to different paradoxes

Referee#2 found this the weakest part of the manuscript. Clearly it needs clarification.

Concerning the Eocene/Oligocene Transition (EOT): The authors thank Referee#2 for bringing to our attention important recent studies of the EOT. We stand in awe of paleoclimate "detectives", such as Lear et al. (2008) and Scher et al. (2011), who deduce details of climate changes that occurred 34 Ma, from seemingly insignificant (to us) benthic foraminifera. These studies confirm that an abrupt climate change, including glaciation of Antarctica, occurred 34 Ma (perhaps in two steps). The modelling paper of Ladant et al. (2014) combines an atmospheric GCM with an ice-sheet model and obtains generally good agreement with proxy data for the EOT. Ladant et al. (2014) state "The reasons for this greenhouse-icehouse transition have long been debated, mainly between the "tectonic- oceanic" hypothesis and the $CO_2$ hypothesis". Our simple EBM weighs in on this important debate, suggesting that both of these forcing factors are necessary to account for the transition. We also support the viewpoint that it was not the polar position of Antarctica but rather the onset of the ACC that led to Antarctic glaciation. The new insight that our EBM brings to the EOT debate is a simple mathematical explanation for the suddenness of the transition. Typically, geological changes occur slowly, over long periods of time. The glaciation of Antarctica was very abrupt on a geological time-scale. Our model suggests a simple mechanism for this sudden transition. Referee#2 is correct in thinking that the timing of the transition in the EBM is in part fortuitous. What is important in our view, is not the prediction of the timing of

transition, but rather the mathematical existence of a sudden transition in the EBM. In our EBM, a sudden transition must occur at some point during the Eocene-Oligocene.

Concerning the "Pliocene paradox": The term "Pliocene paradox" was not introduced by us, but rather by Fedorov et al., Science (2006), who stated "During the early Pliocene, globally averaged temperatures were substantially higher than they are to-day, even though the external factors that determine climate were essential the same". The manuscript authors took this statement as evidence of bistability; that is, two distinct stable equilibrium climate states coexist, with the same forcing factors. In our simple EBM, mathematical analysis confirms the co-existence of two distinct climate states with the same Pliocene forcing factors; one state matches the early Pliocene warm climate and the other is similar to today's climate with a frozen Arctic. Although we can not prove that the much more complex climate of Earth behaves like our simple EBM, we feel that the EBM result gives a new perspective that can be explored further, and has the potential to "solve" the Pliocene paradox stated by Fedorov et al. (2006).

The second "paradox" considered in the manuscript is the abruptness of the climate transition, from equable to frozen, in the Arctic during the Pliocene. The authors agree with Referee#2 and other authors cited that the major cause of the Pliocene transition is $pCO_2$ decrease. However this does not explain why the climate changed drastically, after a long period of gradual change, even though the $pCO_2$ continued to decrease slowly. The EBM provides a mechanism for this sudden change, namely a saddlenode bifurcation, which characteristically causes a "jump" to a new state with only a small change in parameters.

The third paradox concerns the broken North-South symmetry of the Earth's climate, that persisted for 30 Ma, from the EOT to the Pliocene. To paleoclimate scientists, this is not a paradox because it is well understood. To others, it is a surprise to learn that the Antarctic was ice-covered while the Arctic was ice-free for most of this 30 million year time frame. The North-South symmetry was broken by plate tectonics, geography and ocean currents that were very different around the two poles. Knowing this,

it is no surprise that the polar climates were different. Tan et al. (2017) shed light on Arctic glaciation and the closure of the Central American Seaway in the Pliocene. Referee#2 is correct to say that there is nothing fundamentally new in this third paradox of the manuscript. The EBM simply confirms what paleoclimate scientists already know. Therefore, this will be removed from the list of paradoxes in the manuscript and replaced with an observation in the main text.

The fourth paradox is indeed a misunderstanding. The authors were not well-informed on the methodology and efficacy of modern GCM simulations. Our thinking was guided mainly by the pioneering work of Barron, Earth Sci. Rev. (1983), as described in Section 3.2. The Referee is correct to surmise that "the authors mean that . . . some of these models . . . only reach a cold solution and miss another equilibrium." The so-called fourth paradox will be removed from this section and discussed later in the context of the warm equable Cretaceous climate problem. Modern GCM simulations have been successful in closing the gap with Pliocene paleoclimate proxies, and these will be referenced.

Concerning the mid-Cretaceous climate paradox: Barron (Earth Sci. Rev. 1983) stated that proxy data showed mid-Cretaceous climate to be warm and equable, while his GCM simulations gave a climate more like today's climate. He called this discrepancy the warm equable Cretaceous climate problem. In this manuscript, we suggest that both may be correct, if the Earth's climate system (like our EBM) has multiple stable equilibrium states. The actual Cretaceous climate was in a warm equable state, while Barron's GCM computed (correctly) the colder state. Given that the GCM was originally designed to model today's climate, it would not be surprising for it to find the Cretaceous climate state that is more like our own climate. Modern, more successful simulations also will be referenced. The authors agree that the paleogeographic configuration plays an important role. The papers of Donnadieu et al. (2006) and Ladant & Donnadieu (2016) shed light on this issue. There is no "geography" in our EBM; however, the effects of continent areas and sea levels are included indirectly in the ocean

heat transport parameter of this EBM.

Minor comments

In section 2.4 "Positive Feedback Mechanisms", neither ice-albedo feedback nor water vapour feedback is intrinsically new; these have been understood for many years and are included in more sophisticated models. In simple EBM studies, however, such as Payne et al. (2015), ice-albedo feedback is usually modelled as a discontinuous step function and water vapour feedback is usually just a parameter tuned by hand, instead of determined by the Clausius-Clapeyron equation. These changes make the EBM amenable to bifurcation analysis, which had not been done before.

In section 3.2.1, "Permanent El Nino and Hadley cell feedback", the idea that a permanent El Nino in the early Pliocene explains the Pliocene paradox is not supported by us, but has been proposed by others. The idea that Hadley cell feedback contributed to Arctic amplification also is controversial and is not well justified. The papers of Sun et al. Climate Past (2017) and Sun et al. Climate Dynamics (2018) shed light on this Hadley circulation. It is not very relevant to this manuscript and probably is best omitted.

In the Introduction, the authors thank Referee#2 for the revision of the statement beginning "It complements, rather that replaces more detailed . . .".

Ocean dynamics changes are not part of this EBM, except in so far as they change ocean heat transport to the poles. The references for the Pliocene and EOT will be added. Lawrence et al. (2009) and De Schemer (2015) provide important information about SST and ocean dynamics in the Pliocene and Miller et al. (2008) for the EOT.

Conclusion:

Equilibrium Sensitivity Analysis (ESA) for modern day doubling of $CO_2$ will be presented in a follow-up paper on anthropogenic climate change. This analysis will follow the approach of the IPCC.

Glacial/interglacial transitions were driven in large part by orbital forcings that are not included in this EBM. Addition of these issues to the manuscript would significantly lengthen what is already a rather long paper.

Thanks to the suggestions of Referee#2, better and updated insights into the recent bibliography will be added.

The added value of the new EBM simulations in understanding major transitions will be made clear.

---

## Author Response (AR1)

**Author's Response to the Editor and Referees' Reports Dec. 3, 2018**

Manuscript: **An Energy Balance Model for Paleoclimate Transitions**

The authors thank the two reviewers for their constructive criticisms and for pointing out much useful recent work on paleoclimates. We strongly believe the reviewers' suggestions have contributed substantially to improving the manuscript.

5      We have made major modifications to the manuscript based on the reviewers' suggestions and questions. We have previously responded to each of the reviewers' reports, and those responses have been published on-line with the journal. Those responses dealt with the reviewers' issues in a point-by-point manner; we will not repeat those responses here. However, we have made the changes to the manuscript as outlined in those responses. Here we describe the major changes to the manuscript. To begin, we respond to the two major points made by Reviewer 2:

10      *I believe there is room for large improvement in two major directions:*

> 1. *In demonstrating the validation of the EBM for present day and for simple sensitivity experiments as doubling CO2 and glacial/interglacial transition to test the sensitivity of EBM before going to deeper time climate transitions.*
>
> 2. *Depicting a better and updated insight in the recent bibliography concerning each paradox to show more clearly what is the added value of these new simulations in understanding major transitions.*

15      With regard to the first point, we have calibrated the EBM with modern published data and have given this calibration in Appendix A. In addition, we have calculated the Equilibrium Climate Sensitivity (ECS) (the equilibrium surface temperature change due to a doubling in $CO_2$ levels) for our EBM in Appendix B. The calculated value is $3.3°C$, which is in excellent agreement with accepted values published by the IPCC. (In our original response to the reviewers we had indicated we would publish an ECS calculation in a subsequent paper, but since the editor requested specifically that we meet this suggestion of

20 the referee, we have done so in this manuscript. We now agree that it strengthens the validity of EBM.) With regard to the second point, we have expanded the bibliography (about doubling the number of citations), which now includes the many papers pointed out by both reviewers as well as other relevant papers we have found since. We have improved the discussion in the Introduction and throughout the paper by referring to the results of these additional works.

     We believe that the primary contribution of our manuscript is showing that a relatively simple model of the climate that

25 incorporates water vapor feedback, ice albedo feedback, and the contribution of carbon dioxide as a greenhouse gas, has multiple equilibrium states, both warm and frozen, and that rapid transitions (bifurcations) between these states occur due to relatively small changes in the forcing parameters. We suggest that the sudden changes in climate in the past at both poles is a manifestation of this mathematical reality of bifurcations of equilibria. We have tried to highlight this as the primary contribution of this work.

30 ## Detailed Description of Manuscript Changes

     We have provided below the output of "latexdiff", a tool that compares LaTeX source files. The pdf output shows explicitly the alterations made between the original version and the version we are re-submitting at this time. Below we describe these changes. Besides the major changes mentioned below, we made many minor alterations in wording and phrasing in order to improve clarity and flow. We will not comment further on those changes.

35      1. The introduction (Section 1) was substantially complemented with references to other research work. In addition, we have highlighted more clearly the results of this paper. At the end of this section we mention the two new appendices where the model is calibrated and where the ECS value is computed.

     2. Section 2 has been thoroughly re-written. We have moved the determination of numerical values of parameters to Appendix A, so that Section 2 is shorter and focuses on the mathematical model itself. This addresses the one suggestion

40 of the first reviewer to shorten the technical section.

     3. We have altered the EBM model slightly in order to allow for calibration with modern data. In particular, the model now accounts for reflection of sunlight from the atmosphere back into space, and the direct absorption of sunlight by the

atmosphere. This has led to the introduction of two new parameters $\xi_R$ and $\xi_A$, which represent the fraction of incoming solar radiation that is reflected and absorbed, respectively.

4. We have updated Figure 1–3 of Section 2 (old Figures 1,3, and 4) to reflect the modified model.

5. We have changed the definition of the symbol $F_S$ to now be the solar radiation striking the surface, rather than that absorbed by the surface.

6. We have changed how we model the conduction/convection/evapotranspiration of heat from the surface to the atmosphere, $F_C$. The previous manuscript simply had $F_C$ as a constant that we altered depending on the region of the earth to which the model was being applied. However, $F_C$ is certainly dependent on surface temperature, and since we are considering applications where surface temperature changes from below to above freezing it makes sense to model this explicitly. We also found that to calibrate the model to modern data, it was necessary to model $F_C$ as temperature dependent. The precise way we have modelled this dependence is given in Section 2.1, and the parameter values for this functional form are calibrated in Appendix A to modern data.

7. Table 1 has been updated with the new model parameters. A few of the former entries in the table, that were only used as intermediate values, have been removed in order to keep the table to one page. All values for the model parameters are still present in the Table.

8. In order to calibrate the model to modern data, we have also added absorption of infrared radiation due to liquid and solid water in clouds. The previous model ignored this contribution. However, cloud absorption constitutes about 25% of total absorption and so to calibrate the model with published energy balance values, it was necessary to include this contribution. Since there is virtually no information on paleoclimate cloud cover, we have kept this value as a constant even though it clearly will depend on surface temperature, surface topography, and latitude. We did try modelling the dependence of cloud absorption on surface temperature, like for $F_C$, but found no qualitative change in our results and so have left cloud absorption as a global and temporal constant contribution.

9. Section 3.1 and Section 3.2 have been swapped in the new manuscript so that the Pliocene Paradox is treated first.

10. Figures 4–8 in Section 3 (old Figures 5–9) have been re-drawn to correspond to the modified model. The values of the atmospheric and ocean heat transport forcings, $F_A$ and $F_O$, are modified slightly from the previous model in order to compensate for the model changes introduced. The qualitative features of these figures have not changed. In particular, the bifurcations remain present, where warm equilibria disappear in a saddle-node.

11. Section 4, the conclusion, has had a few minor alterations.

12. Appendices A and B are new. Appendix A gives the empirical calibration of the EBM model to published modern data. Some of this material is new, corresponding to new aspects of the modified model like atmospheric reflection and absorption of sunlight, and some of the material has been moved here from the old Section 2, so that all numerical calibration is in one place.

13. Figure A1 was Figure 2b in the old version. Figure A2 is a new figure depicting the dependence of $F_C$ on temperature.

14. Appendix B gives the equilibrium climate sensitivity (ECS) calculation for the EBM. It predicts an ECS value of $3.3°C$ for a doubling of $CO_2$ concentration from 270 ppm to 540 ppm. Figure B1 is new, depicting the ECS.

15. The bibliography has increased from 53 to 107 entries.

Again, we thank the reviewers for their very thoughtful and detailed reviews. We believe that the revised manuscript addresses all of the issues raised by them and the editor. We respectfully re-submit the manuscript for publication.

Sincerely,

Brady Dormans, William F. Langford, and Allan R. Willms

[revised manuscript text omitted]

---

## Referee Report (RR1)

**Second review on the manuscript entitled "An Energy Balance Model for Paleoclimate Transitions" by Brady Dortmans, William F. Langford, and Allan R. Willms.**

In the first round I raised major comments on this paper which was promising but suffered from two major caveats. One concerned the model calibration and the other one referred to the lack of clarity on original findings of this manuscript compared with important literature already published and sometimes not even cited.

In this new manuscript the authors seriously answered my major comments.

1 they added appendices concerning comparison of model results and climatology for present day and also investigated the sensitivity to a $CO_2$ doubling

2 They largely increased the number of references cited which is now very important but, since they are using their EBM model in very different paleoclimatic contexts, this justifies such a huge number of references.

Moreover, they clarify the originality but also the limits of their study (page 4).

Please find below my comments to the authors responses and to the new manuscript

1 General introduction: The papers I cited concerning illustration of quaternary bifurcations are to my point of view very useful and relevant to convince Climate of the Past readers of the interest of simple models and bifurcations in a much better constrained context with many climatic proxies than model of intermediate complexity as EBM. Indeed, for instance Paillard and Ganowpolski papers demonstrated that bifurcations may explain abrupt transitions.

Conversely, in Stap 2017, paleogeography is poorly constrained through time and even if the periods are more relevant, the demonstration is, for me, less convincing. Indeed between 38 and 5 Ma, which is the period under consideration in Stap 2017, many forcing factors have changed, as for example paleography, including Tethys shrinkage, Tibetan plateau uplifts, opening and closing of seaways that are not accounted for in this paper.

I have a comment concerning the author statement " However, they would be very interested in exploring with GCM experts the possibility of multiple stable equilibrium states in a paleoclimate GCM".
A good illustration concerning the comparison of EMIC and GCM and the existence of bifurcation is provided in the context of mid Holocene green Sahara. Whereas bifurcation is depicted by Clausen (Claussen et al. GRL 1999, Simulation of an abrupt change in Saharan vegetation in the Mid-Holocene) using an EMIC, according to Liu et al ( Z. Liu et al. Science 2009 Transient Simulation of Last Deglaciation with a New Mechanism for Bølling-Allerød Warming) , such a bifurcation is not necessary when using a transient GCM simulation.

I made this comment because I think the authors should discuss more the competition between EMIC showing bifurcation and transient GCM simulations that include non linear processes and not necessarily showing bifurcations.

Page 4, the authors present carefully the limits and the possibility of their new EBM, but once again, considering the progress made on coupling OAGCM coupled with ISM and describing the triggering of Antarctica (Ladant 2015) and Greenland (recently published Tan et al 2018), we can wonder if the concept of bifurcation is necessary. The authors should justify the interest of the EBM in these 2 different contexts.

My opinion as already mentioned in the previous review is that a hierarchy of models is necessary. GCM explored only some trajectories while EMIC may explore a much larger range of parameters.

2 Model presentation and publication strategy

The authors accounted for my previous claim concerning the necessary validation of their model and its sensitivity. Nevertheless, I am still puzzled that the authors provide a first manuscript devoted to the use of the new model for large climate change periods and claim that they will, in a second paper, describe and validate the model and use it for more constrained sensitivity experiments for present day and future climates. I would have preferred in a first paper a solid description of the strong and weak points of the new EBM and in a second paper, applications to key periods of deep time climate changes. Moreover the detailed description of the EBM characteristics provided till page 19 is still difficult to follow. How far EBM parameterizations favor bifurcations ? Moreover such a detailed description of the tool would require some insights on validation of mean present day climate and its sensitivity.

Nevertheless the minimum knowledge has now been provided in the appendices.

Concerning the first appendix, it is devoted on modeling tuning to reproduce present day climate. Still an important figure is missing showing for instance the annual present day temperatures simulated by the model compared to a climatology data set.

3 Paradoxes

The authors have deeply modified this section. Nevertheless, new papers have been published that, at least for Pliocene paradox should be included and discussed. Indeed, papers depicting the transition from a Northern Hemisphere free of ice caps to Greenland inception using OAGCM asynchronously coupled with sophisticated ice sheet models as for example Tan et al Nat. Com 2018 and Willeit QSR 2015 partly solved the "Pliocene paradox". Therefore, this concept should be revisited.

More generally, Baron series of numerical experiments did not account for many processes that are nowadays better simulated by coupling AGCM with ocean, vegetation and ice sheets. Therefore, also the mid-Cretaceous paradox is nowadays not so clear. But still the issue of reproducing flat equator to pole thermal gradient is a challenge.

.

---

## Author Response (AR2)

Author's Response to Second Referee Report:
CP-2018-56-referee-report-1.pdf

Feb 23, 2019

The authors thank this anonymous referee for his/her many helpful suggestions and constructive criticisms. The Referee's advice has lead to significant improvements of the paper. The authors are mathematicians and are not as familiar with the climate science literature as the Referee.

The authors have incorporated the referee's suggestions, most notably in the introduction and in the applications of the model. The marked-up manuscript, produced with software Latexdiff, highlights these changes. There are no changes in the mathematics or the figures, as compared to the December version of the manuscript. That earlier revision added a comparison of the model with modern climate data and a determination of the Equilibrium Climate Sensitivity (ECS) of the model.

The Referee is justifiably puzzled that the authors chose to apply their model to paleoclimates first and only later to present and future climates. This is perhaps the reverse of normal practice. The reason for this approach is that the authors' primary interest is to explore the possible roles that "bifurcation theory" may play in climate change. We know that in nonlinear mathematics models, bifurcations can cause abrupt change in the state of a system, driving it to a new and dramatically different state. Other authors have found bifurcations in climate models, but our analysis seeks a deeper understanding of such bifurcations. Our preliminary work on anthropogenically-driven climate change suggests that there will be a bifurcation in our future, if human production of CO2 does not decrease. However, this forecast would carry little weight with an unvalidated model. In reviewing past climate changes, we selected the EOT and the Pliocene as possible examples, where bifurcations may have played an important role in abrupt climate change. Our model reproduces (and perhaps explains)

these sudden changes. In effect, we are using these past climate transitions to validate our model for predictions of the future.  The revised manuscript attempts to make this point clearer, in the Introduction.

Finally, the Referee asks for an additional figure, "showing for instance the annual present day temperatures simulated by the model compared to a climatology data set".  The authors feel that this issue was addressed adequately in the two Appendices added to the paper in the December Revision. Most models can be adjusted to agree with a limited data set; however, we are not concerned primarily with precise agreement between our model and hard data.  Our interest is in the qualitative nature of change, whether smooth or abrupt, and whether bifurcation theory may be used to help explain and even predict abrupt climate changes, past and future.

William F. Langford
For Allan Willms and Brady Dortmans

**An Energy Balance Model for Paleoclimate Transitions**

Brady Dortmans, William F. Langford, and Allan R. Willms

Department of Mathematics and Statistics, University of Guelph, 50 Stone Road West, Guelph, ON, Canada N1G 2W1

*Correspondence to:* W. F. Langford (wlangfor@uoguelph.ca)

**Abstract.** A new energy balance model (EBM) is presented and is used to study Paleoclimate transitions. While most previous EBMs dealt only with the globally averaged climate, this new EBM has three variants: Arctic, Antarctic and Tropical climates. The EBM incorporates the greenhouse warming effects of both carbon dioxide and water vapour, and also includes ice-albedo feedback and evapotranspiration. The main conclusion to be inferred from this EBM is that the climate system may possess multiple equilibrium states, both warm and frozen, which coexist mathematically. While the actual climate can exist in only one of these states at any given time, the EBM suggests that climate can undergo transitions between the states, via mathematical saddlenode bifurcations. This paper proposes that such bifurcations have actually occurred in Paleoclimate transitions. The EBM is applied to the study of the *Pliocene Paradox*, the *Glaciation of Antarctica* and the so-called *warm, equable climate problem* of both the mid-Cretaceous Period and the Eocene Epoch. In all cases, the EBM is in qualitative agreement with the geological record.

*Copyright statement.* TEXT

**1 Introduction**

For approximately 75% of the last 540 million years of the paleoclimate history of the Earth, the climate of both polar regions was mild and free of permanent ice-caps  (Cronin, 2010; Crowley, 2000; Hubert et al., . Today, both North and South Poles are ice-capped; however, there is overwhelming evidence that these polar ice-caps are melting. The Arctic is warming faster than any other region on Earth. The formation of the present-day Arctic and Antarctic ice-caps occurred abruptly, at widely separated times in the geological history of the Earth. This paper explores some of the the underlying mechanisms and forcing factors that have caused climate transitions in the past, with a focus on transitions that were *abrupt*. The understanding gained here will be applied in a subsequent paper to the important problem of  determining whether anthropogenic climate change may lead to an abrupt transition in the future.

We present a new two-layer energy balance model (EBM) for the climate of the Earth. General knowledge of climate and of climate change has been advanced by many studies employing simple EBMs  (Budyko, 1968; Kaper and Engler, 2013; McGehee and Lehman, 2012; North et al., 1981; Payne et al., 2015; Sagan and Mullen, 1972; S . In general, these EBMs facilitate exploration of the relationship between specific climate forcing mechanisms and the resulting

climate changes. The EBM presented here includes a more accurate representation of the role of greenhouse gases in climate change than has been the case for previous EBMs. The model is based on fundamental principles of atmospheric physics, such as the Beer-Lambert Law, the Stefan-Boltzmann Law, the Clausius-Clapeyron equation and the ideal gas equation. In particular, the modelling of water vapour acting as a greenhouse gas in the atmosphere, presented in Subsection 2.3.3, is more physically accurate that in previous EBMs and it shows that *water vapour feedback* is important in climate change. Also, *ice-albedo feedback* plays a central role in this EBM. The nonlinearity of this EBM leads to *bistability* (existence of multiple stable equilibrium states), to *hysteresis* (the climate state realized in the model depends on the past history) and to *bifurcations* (abrupt transitions from one state to another). Previous climate models exhibiting multiple equilibrium states have been studied for example by North et al. (1981); Paillard (1998); Ferreira et al. (2010); Thorndike (2012); Stap et al. (2017). 
[revised manuscript text omitted]